# “In Less than No Time”: Feasibility of Rotational Thromboelastometry to Detect Anticoagulant Drugs Activity and to Guide Reversal Therapy

**DOI:** 10.3390/jcm11051407

**Published:** 2022-03-04

**Authors:** Vittorio Pavoni, Lara Gianesello, Duccio Conti, Piercarlo Ballo, Pietro Dattolo, Domenico Prisco, Klaus Görlinger

**Affiliations:** 1Anesthesia and Intensive Care Unit, Emergency Department and Critical Care Area, Santa Maria Annunziata Hospital, Bagno a Ripoli, 50012 Florence, Italy; vittorio.pavoni@uslcentro.toscana.it (V.P.); duccio.conti@uslcentro.toscana.it (D.C.); 2Department of Anesthesia and Intensive Care, Orthopedic Anesthesia, University-Hospital Careggi, 50134 Florence, Italy; 3Cardiology Unit, Santa Maria Annunziata Hospital, Bagno a Ripoli, 50012 Florence, Italy; piercarlo.ballo@uslcentro.toscana.it; 4Nephrology Unit Florence 1, Santa Maria Annunziata Hospital, Bagno a Ripoli, 50012 Florence, Italy; pietro.dattolo@uslcentro.toscana.it; 5Department of Experimental and Clinical Medicine, University of Florence, 50134 Florence, Italy; domenico.prisco@unifi.it; 6Department of Anesthesiology and Intensive Care Medicine, University Hospital Essen, University Duisburg-Essen, 45147 Essen, Germany; kgoerlinger@werfen.com; 7Medical Department, Tem Innovations, 81829 Munich, Germany

**Keywords:** bleeding, thromboembolic events, thromboprophylaxis, oral anticoagulants, heparins, rotational thromboelastometry

## Abstract

Anticoagulant drugs (i.e., unfractionated heparin, low-molecular-weight heparins, vitamin K antagonists, and direct oral anticoagulants) are widely employed in preventing and treating venous thromboembolism (VTE), in preventing arterial thromboembolism in nonvalvular atrial fibrillation (NVAF), and in treating acute coronary diseases early. In certain situations, such as bleeding, urgent invasive procedures, and surgical settings, the evaluation of anticoagulant levels and the monitoring of reversal therapy appear essential. Standard coagulation tests (i.e., activated partial thromboplastin time (aPTT) and prothrombin time (PT)) can be normal, and the turnaround time can be long. While the role of viscoelastic hemostatic assays (VHAs), such as rotational thromboelastometry (ROTEM), has successfully increased over the years in the management of bleeding and thrombotic complications, its usefulness in detecting anticoagulants and their reversal still appears unclear.

## 1. Introduction

Anticoagulant drugs are recommended for prophylaxis against venous thromboembolism (VTE) in high-risk patients and for the treatment of VTE [1,2]. They are also indicated for other medical conditions including nonvalvular atrial fibrillation (NVAF) and early treatment of patients with acute myocardial infarction and unstable angina [3]. Moreover, they are used to prevent clotting during dialysis and clinical procedures [3].

In certain circumstances, such as in patients presenting traumatic injuries with severe bleeding or when immediate invasive or surgical interventions are required, to assess the level of anticoagulation and monitoring, reversal therapy may be essential.

During vitamin K antagonist (VKA) treatment, standard coagulation tests (i.e., prothrombin time (PT)) provide information on the level of anticoagulation [4]; however, in urgent clinical scenarios, the turnaround time can be long.

Heparin treatment is usually monitored by activated partial thromboplastin time (aPTT); however, the responsiveness of the laboratory reagents used in aPTT tests can vary widely [5].

LMWHs prolong the aPTT with a degree dependent on reagent sensitivity and plasma concentration [6]; the chromogenic antifactor Xa (FXa) assay is the only one available for their monitoring [7].

In regard to treatment with direct oral anticoagulants (DOACs), although they do not require laboratory monitoring, recent guidelines on the management of major bleeding following trauma have suggested the measurement of calibrated anti-FXa activity in patients treated with oral direct anti-FXa agents and the estimation of dabigatran plasma levels using diluted thrombin time [8]; however, in urgent conditions, the availability of these specific assays may be limited.

In clinical practice, the need for accurate and prompt coagulation information has created an interest in viscoelastic tests (VETs). Rotational thromboelastometry (ROTEM, Tem Innovations, Munich, Germany) is a global hemostasis assay based on the viscoelastic properties of blood capable of assessing coagulation function, platelets, and fibrinogen contribution to clot formation and fibrinolytic components [9,10,11]. This device has been utilized in trauma and surgical and critical care as an adjunct to conventional coagulation tests to guide resuscitation and transfusion strategies and to improve patient outcomes [12]. In addition, it has been used in the diagnosis of bleeding conditions due to quantitative and qualitative as well as hereditary and acquired fibrinogen disorders [13]. Furthermore, the turnaround time for results of ROTEM has been shown to be significantly shorter than those of conventional laboratory tests [14]. 

To date, the role of a ROTEM device for coagulation monitoring in anticoagulated patients is unclear. The purpose of this narrative review is to analyze the literature on the ability of ROTEM to detect different anticoagulant drugs and to monitor their effects and reversal when an invasive procedure or urgent surgery would be more efficiently and safely dealt with due to a rapid evaluation of the coagulation system. 

The mean characteristics of anticoagulant drugs considered in the present review are shown in Table 1.

## 2. ROTEM Device

The ROTEM system is a whole blood viscoelastic hemostasis analyzer that evolved from the original thromboelastography (TEG) system introduced by Hellmut Hartert in 1948 [15]. It is based on the concept of the “shear modulus,” a measure of the clot that represents the tendency to deform due to the action of opposing forces. 

In ROTEM a blood sample is injected in a cylindrical sample cup fixed in a temperature-adjusted metal cup holder and in an attached disposable pin which moves through an angle of 4.75° every 5 s. As long as the coagulation process continues, the device detects the variation of clot strength between the pin and the cup wall.

The ROTEM system includes the semiautomated ROTEM delta that works with a computer-driven automated pipette and provides four independent channels for VET and the new-version ROTEM sigma that is a cartridge-based fully automated closed thromboelastometry system. The recent sigma model [16] has several advantages compared with the older delta, such as minimizing the manipulation of the blood sample and obviating the need for semiautomated pipetting. 

ROTEM explores the coagulation pathway using citrated whole blood (300 µL per assay), which is recalcified and activated by tissue factor (TF) (extrinsic pathway), ellagic acid (intrinsic pathway), or ecarin (direct prothrombin activation). Recalcifying the citrated whole blood in the cup, with the addition of activators, determines the formation of clot strands between the pin and the cup wall, which impairs the pin rotation. These changes in pin movement are detected by a LED light-mirror-light detector system, and the signal is processed and transformed into a thromboelastometric curve (temogram). The parameters obtained in ROTEM and their meaning are shown in Figure 1.

In both ROTEM versions (delta and sigma), extrinsically activated assays (EXTEM, FIBTEM, and APTEM) and intrinsically activated assays (INTEM and HEPTEM) are available; ecarin-activated assay (ECATEM) and two nonactivated assays (NATEM and NA-HEPTEM) are disposable in the delta version as optional assays. EXTEM assay uses TF as an activator and is analogous to PT. INTEM is similar to aPTT and uses phospholipid and ellagic acid as activators. FIBTEM uses in addition to the EXTEM reagent a platelet inhibitor (cytochalasin D) that blocks the platelet contribution to clot formation, leaving the fibrin contribution to clot firmness. APTEM uses in addition to the EXTEM reagent aprotinin to inhibit fibrinolytic proteins; normalization of lysis parameters in APTEM compared with EXTEM or FIBTEM suggests the presence of hyperfibrinolysis. All extrinsically activated ROTEM assays (EXTEM, FIBTEM, and APTEM) contain polybrene to neutralize up to 5 IU/mL unfractionated heparin. HEPTEM assay is virtually identical to INTEM assay, but with the addition of heparinase to neutralize up to 7 IU/mL unfractionated heparin [11]. ECATEM is an assay that uses ecarin to initiate the coagulation cascade at the step of thrombin generation. Ecarin is a prothrombin activator derived from the venom of the saw-scaled viper *Echis carinatus*. ECATEM is sensitive for direct thrombin inhibitors (e.g., argatroban, bivalirudin, dabigatran), but it is not sensitive to heparin [17,18]. Instead, the native thromboelastometric test (NATEM), a nonactivated assay, is used to assess whole blood clot formation without the addition of coagulant activators other than recalcification and spontaneous contact activation. The NATEM mode enables measurement under near-physiological conditions. The test is increasingly used as a sensitive tool for the evaluation of the endogenous activation of hemostasis [19,20]. It is longer than activated tests but is very sensitive to any endogenous activator, such as TF expression on circulating monocytes in infection, liver cirrhosis, or malignancies [21,22,23], and to subsequent thrombin generation. However, only small data exist about stability and reproducibility over time. Meesters et al. [24] described a time-dependent effect on NATEM clotting time (CT) and clot formation time (CFT) after 90 min of blood storage and recommends analysis at a standardized time point, preferably directly after withdrawal of citrated blood. By adding heparinase to this, creating the native thromboelastometric (NA-HEPTEM) assay that contains heparinase in addition to CaCl_2_ that eliminates up to 7 IU/mL heparin, it is possible to abolish a potential heparin effect. 

Finally, a modified ROTEM test, called PiCT (prothrombinase-induced clotting time, Pentapharm, Basel, Switzerland)-ROTEM test, is a clotting assay sensitive to FXa and factor IIa (FIIa) inhibitors; it is based on the addition of FXa and snake venom RVV-V (Russell’s viper venom factor V activator) specifically activating factor V and phospholipids to platelet-poor plasma [25]. It has been demonstrated that the PiCT reagent allows the determination of FXa inhibition in plasma [26]. 

## 3. Parenteral Anticoagulation

### 3.1. Unfractionated Heparin

Unfractionated heparin (UFH) consists of a heterogeneous glycosaminoglycan mixture with a mean molecular weight (MW) of 15 kilodaltons (kDa), ranging from 3 to 30 kDa. Heparin binds to antithrombin, resulting in a conformational change in the latter. The heparin antithrombin complex inactivates FXa, a key enzyme present at the beginning of the common pathway of the coagulation cascade. Additionally, heparin simultaneously binds to thrombin with an inhibitory effect on this enzyme. Heparin has an equivalent anti-FXa and anti-FIIa activity (ratio close to 1) [27]. 

Several tests are available to monitor UFH therapy, including whole blood activated clotting time (ACT), aPTT, and plasma heparin concentration measured by anti-FXa activity. 

The aPTT is the most widely used test to determine the degree of anticoagulation with UFH both when usual therapeutic doses are used and after reversal with protamine. However, apart from all the difficulties regarding standardization of the test and the influence of lupus anticoagulant and various coagulation factor deficiencies, the weak correlation between UFH levels and aPTT has been demonstrated [28]. Furthermore, in patients with so-called heparin resistance, false-low aPTT values (despite adequate anti-FXa activity) may trigger UFH dose escalations, thus increasing the risk for bleeding [29]. 

Anti-FXa chromogenic substrate assays [29] may provide a better and more reliable anticoagulation monitoring of UFH, although it may cause an unavoidable delay between the blood sampling and the completion of the laboratory report. Furthermore, they are not equivalent to all, and they result in increased cost due to the need for specialized instrumentation for test and quality control samples [30]. 

Regarding point-of-care devices, measurement of the ACT was first described by Hattersley [31] and later implemented clinically by Bull and colleagues [32]; it has become the gold standard for anticoagulation management during cardiopulmonary bypass (CPB). However, it remains controversial whether ACT itself is a valid measure of heparin anticoagulation because of multiple clinical factors that affect results (e.g., hemodilution, excess of protamine, low platelet numbers, and low fibrinogen levels) [33]. Furthermore, in [34] ACT values appeared not to be a sensitive indicator of low heparin concentrations present during CPB after reversal with protamine.

In clinical settings, ROTEM have been reported as a technique for monitoring heparinized patients during CPB [35,36] and in critically ill patients undergoing UFH anticoagulation [37,38]. Ortmann et al. [35] demonstrated that a ROTEM device can be used during CPB to assess a coagulation system. Gronchi et al. [36] in a prospective observational study—in 20 patients undergoing coronary artery bypass grafting—validated EXTEM and HEPTEM assays; the authors demonstrated that the administration of heparin 300 IU/kg (very high concentration) induced a significant change in all EXTEM and HEPTEM parameters except for MCF and that all parameters remained stable after the protamine administration. 

Moreover, for what concerns the correlation between laboratory measurements and ROTEM parameters during CPB, Ortmann et al. [35] reported a good correlation of INTEM clotting time (CT) with aPTT measurement (r = 0.65, 95% CI 0.45–0.78) and a moderate correlation for EXTEM-CT and INR from the laboratory (r = 0.58, 95% CI 0.36–0.73). Similar results were found in an observational study in ICU patients [37]; the authors demonstrated a significant correlation between aPTT and the changes of ROTEM-CT with heparin dose or heparin infusion rate, but no correlation between clot formation time (CFT) and maximum clot firmness (MCF) and heparin dose or infusion rate. On the contrary, Prakash et al. [38], in 20 critically ill patients treated with extracorporeal membrane oxygenation (ECMO), demonstrated a weak correlation between INTEM-CT and aPTT results during UFH anticoagulation (r = 0.31, *p* < 0.001, 95% CI (0.17, 0.43)) and no correlation between aPTT and INTEM-CFT (r = 0.06, *p* = 0.04, 95% CI (−0.19, 0.08)). INTEM-CT and ACT were less sensitive than aPTT to UFH therapy. Despite achieving therapeutic aPTT targets, the majority (>50%) of INTEM-CT results were within normal limits; INTEM-CT was affected only by high heparin infusion rates. However, this study presented the same limitations: the HEPTEM assay was not performed, limiting the evaluation of the specific effect of heparin on INTEM-CT; moreover, INTEM-CT was compared with aPTT, which is itself profoundly affected by various disturbances in coagulation (e.g., hemodilution, acquired Von Willebrand disease) [39,40] induced by extracorporeal circuit. 

Different studies demonstrated that aPTT and ACT are not correlated with the presence of heparin in the plasma during extracorporeal support [41,42,43]. On the contrary, other studies showed a good correlation between INTEM/HEPTEM CT ratio and heparin values > 0.1 IU/mL [44,45,46]. Particularly, Mittermayr et al. [45] demonstrated that the INTEM/HEPTEM CT ratio correctly identified 56 of the 58 samples as not containing residual heparin and correctly detected residual heparin in 3 of the only 6 samples showing elevated anti-FXa values (>0.1 IU/mL) after CPB. 

Ichikawa et al. [44] demonstrated that INTEM/HEPTEM CT ratio correlated well with heparin concentration after weaning from CPB and heparin reversal by protamine (r = 0.72), whereas aPTT (r = 0.36) and ACT (r = 0.12) did not. 

Recently, Hanke et al. [47] found in an in vitro study that ROTEM enables the detection of the effects of heparin and protamine on coagulation. In this study, the CT was significantly prolonged in INTEM (*p* < 0.0001), but not in HEPTEM, whereas overdosage of protamine determined an increase in the CT in both tests (INTEM and HEPTEM); the authors suggested that calculating the INTEM/HEPTEM CT ratio, which increases with heparin excess but remains at 1 with protamine excess, it can distinguish impairment of coagulation caused by heparin or protamine.

Furthermore, Schaden et al. [48] showed in an in vitro study a linear correlation between UFH levels and CT in the PiCT-ROTEM assay with an excellent correlation (Spearman’s rank correlation coefficient = 0.92, *p* < 0.01). 

In conclusion, from analysis of the literature, ROTEM might be useful in detecting heparin with high sensitivity and in excluding residual heparin after CPB in cases showing prolonged ACT or increased bleeding after protamine administration. In particular, the INTEM/HEPTEM CT ratio appears to be a good predictor of heparin rebound, superior to ACT and aPTT values.

### 3.2. Low-Molecular-Weight Heparins

Low-molecular-weight heparins (LMWHs) are a class of drugs derived from UFH. Each LMWH has its own spectrum of biochemical properties, determined by the characteristics of the parent UFH and by the effects of the cleavage process on the structures and functions of the low-molecular-weight fragments [26,49]. The variation in the anticoagulant potency of LMWHs depends on different inhibition degrees of the coagulation FXa and FIIa. LMWHs, such as enoxaparin (mean MW of 4.5 kDa) and dalteparin (mean MW of 5 kDa), have, to varying degrees, relatively greater anti-FXa activity than antithrombin activity; heparin molecules of MW less than 5.4 kDa are capable of catalyzing only FXa inactivation [50]. 

Due to the more predictable pharmacokinetic and pharmacodynamic proprieties than UFH, LMWHs are widely used in clinical practice. A peak therapeutic effect with LMWH occurs 3 to 5 h after administration, with half-lives of 4 to 5 h [51]. Both renal failure and advancing age are known to prolong its half-life and delay elimination [52]. 

A major advantage of LMWHs is that they do not require monitoring during the therapy; however, in the specific clinical settings in whom pharmacokinetics may be altered by both comorbid conditions and severity of illness [53,54], in surgical scenarios where the bleeding risk requires evaluation of anticoagulant intensity [55], or in the perioperative management of hypofibrinogenemic patients in which clinicians should take into account both the bleeding and thrombotic risk, monitoring may be required to avoid over- or underanticoagulation [56]. 

The aPTT is not useful in monitoring LMWH because the anticoagulant effect is produced mainly through the inhibition of FXa, and the prolongation of the aPTT is largely dependent on low thrombin activity [57,58]. 

The anti-FXa assay method using a chromogenic substrate is currently the gold standard for monitoring LMWH therapy [7]. However, in many hospitals this laboratory test is not available, and when available, the turnaround can be long. Furthermore, the clinical application appears uncertain [59]. Some studies have demonstrated a poor correlation between anti-FXa activity and anticoagulant effects of LMWHs [54,60]; this could depend on the poor comparability between commercially available anti-FXa chromogenic assays [61], the timing of blood sampling because the plasma concentration is not constant during the period of two injections [62], and the different pharmacodynamic pattern of LMWHs [63]. Furthermore, in patients with antiphospholipid syndrome, the presence of antiphospholipid antibodies has been shown to falsely raise anti-FXa levels due to a direct inhibition of specific FXa activity in a chromogenic assay [64]; in addiction, due to cytokine release syndrome or propofol use, many critically ill patients also develop hypertriglyceridemia, which has also been shown to falsely increase anti-FXa levels [65]. Moreover, the determination of anti-FXa levels is only validated for values higher than 0.1 U/mL; concentrations below this limit are not specified [66]. Measurement of anti-FXa activity in plasma represents only one aspect of LMWH’s activity. Dosage of anti-FXa activity is performed in platelet-poor plasma (PPP). However, platelet factor 4 released from activated platelets partially neutralizes the anticoagulant activity of LMWHs, although to a lesser extent when compared with UFH [67]; this leads to partial inhibition assays of LMWHs’ antithrombotic activity [68]. 

The ROTEM system could reflect the antithrombotic efficacy of heparins or even predict the bleeding risk. In the literature, various studies have been conducted to evaluate the usefulness of a ROTEM device in the detection of parenteral anticoagulant drugs (Appendix A) [69,70,71,72,73,74,75,76,77,78,79,80,81,82,83,84]. One study [69] evaluated in vitro whether two types of LMWH, enoxaparin and tinzaparin, added in different concentrations, had dose-dependent effects on INTEM-CT. The authors found that the ROTEM-CT was prolonged by LMWH at 0.5, 1.0, and 1.5 anti-FXa IU/mL concentrations in a significant dose-dependent manner. Furthermore, equivalent doses of tinzaparin prolonged CT more than enoxaparin (*p* < 0.05). Probably, this result could be explained by the fact that tinzaparin with a mean of MW 6.8 kDa is more similar to UFH and inhibits FXa only twice as strongly as FIIa [70]; notably, there is also evidence that UFH and LMWH with larger MW (>2 kDa) exert an anticoagulant effect through plasma TF pathway inhibitor [71]. Similarly, Feuring et al. [72], using EXTEM and INTEM reagents, demonstrated a dose-dependent prolongation of CT and CFT with 1 or 10 µg/mL dalteparin, corresponding to an anti-FXa activity of 100 and 1000 IU, respectively. This in vitro study revealed that the ROTEM device was able to detect anticoagulant activities of dalteparin only at considerably increased anti-FXa levels, which are well beyond therapeutic levels. In the same line, in women with thrombophilia receiving LMWH at the prophylactic dose, the use of ROTEM revealed unchanged mean CTs for INTEM and EXTEM assays before and during each trimester of pregnancy. However, the authors assumed that the lack of changes in CFT in the presence of hypercoagulability during pregnancy may reflect the influence of LMWH on coagulation [73].

The impact of LMWH dosage on ROTEM assays has been demonstrated by other studies [74,75,76]. Cvirn et al. [74] showed that the addition of nadroparin or enoxaparin (0–1.0 anti-FXa IU/mL) determined a dose-dependent prolongation of CT and CFT and a dose-dependent reduction in MCF and α angle. Notably, Jilma-Stohlawetz et al. [75] found an increase in CT by 70%, and CFT doubled at a concentration of 0.2 IU/mL enoxaparin; moreover, a concentration of 0.4 IU/mL prolonged the CT by 140% and CFT by 130% (*p* < 0.01).

On the other hand, Christensen et al. [76] found a longer INTEM-CT on the first postoperative day in patients undergoing video-assisted thoracoscopic surgery and receiving dalteparin at the thromboprophylaxis level (5000 IU once daily) compared with patients without dalteparin (INTEM-CT 167 (19) vs. 156 (14), *p* = 0.01), but no differences in the coagulation profile. Notably, in 46 pregnant women with a history of unprovoked or estrogen-associated VTE, Stanciakova et al. [77] prospectively observed a prolongation of EXTEM-CT at the 16th–17th week of pregnancy caused by the onset of the effect of LMWH after its repetitive administration in the course of pregnancy; after adjustment of LMWH at the 26th–28th week, EXTEM-CT at the 35th–36th week was prolonged, probably due to the effect of the increased dose of LMWH. Moreover, based on the ROTEM results that showed a shortening of INTEM-CT during pregnancy and a decrease in HEPTEM-CT at the 26th–28 week, the authors recommended an increase in LMWH dose mostly at the 26th–28th week of the pregnancy. Therefore, they concluded that ROTEM can be a useful tool for the individualized optimization of thromboprophylaxis in high-risk pregnant patients.

Regarding the effects of LMWHs on clot stability, Thomas et al. [69] reported that both enoxaparin and tinzaparin treatment had no effect on MCF; the authors observed no significant correlation between the concentration of LMWH and clot strength (MCF and CFT). These results were in contrast with a previous in vitro study conducted by Feuring et al. [72], in which ROTEM-MCF was affected by supratherapeutic levels of dalteparin, although not therapeutic levels. Notably, MCF is not influenced by the inhibition of the coagulation process induced by LMWH but is mainly affected by fibrinogen concentration, platelet number, and platelet contractile forces [78,79]. In the same line, Gerotziafas et al. [80] found that therapeutic doses of enoxaparin affected thromboelastography maximum amplitude (TEG-MA) (it corresponds to ROTEM-MCF) in healthy volunteers. LMWH prevents clot initiation but not propagation or firmness; this does not necessarily mean that LMWH does not affect clot propagation in vivo, since both the above-mentioned studies are ex vivo [72,80] in which the authors used monitoring of coagulation in a stagnant container that can contribute to the cell-mediated positive-feedback loops that maintain propagation.

The kinetics of clot formation is strongly influenced by the initial trigger of coagulation (i.e., activation of TF pathway or intrinsic pathway) and by the concentration of TF. Sorensen et al. [81] introduced a thromboelastometric test modification using a low concentration of TF in order to enhance the sensitivity for hemostatic defects. Low-concentration TF, as a trigger of blood coagulation, has been an effective physiological stimulus [82]. Notably, low-TF activated ROTEM has been demonstrated to be sensitive for the detection of enoxaparin [83]. In patients undergoing carotid angioplasty receiving postoperative nadroparin at the prophylactic dose, minimal TF-triggered whole blood thromboelastogram resulted in significant longer CT and CFT and lower α angle compared with either INTEM or EXTEM assay. Moreover, prolongation of CFT of a low-TF activated ROTEM was significantly correlated to the levels of anti-FXa activity (*p* = 0.04; r^2^ = 0.7). Conversely, nadroparin had no effect on MCF, assessed by either low-TF activated ROTEM or INTEM assay. 

Recently, Örlander et al. [84], using citrated whole blood from 15 critically ill patients to which different concentrations of enoxaparin had been added in vitro, found a prolongation of ROTEM-CT and CFT in both NATEM and INTEM with increasing in enoxaparin between 0.4 and 0.6 IU/mL anti-FXa concentration. Moreover, the NATEM/HEPTEM-CT ratio seems to be very sensitive to LMWH doses.

An in vitro experiment [66] evaluated enoxaparin determination in whole blood with ROTEM using specific test modifications, including prothrombinase-induced clotting time (PiCT), low-TF activation, and heparinase-dependent tests. This study demonstrated a strong correlation between enoxaparin anti-FXa concentrations and PiCT-ROTEM (*p* < 0.01) and Low-TF-ROTEM (*p* < 0.01). Notably, the CT in PiCT-ROTEM showed a very high correlation to anti-FXa concentrations, particularly in the low-concentration range. Practical test performance of PiCT-ROTEM was more convenient that the Low-TF- ROTEM.

In conclusion, there is currently a paucity of data to recommend a standardized protocol for the use of ROTEM in LMWH therapy monitoring. However, since the inhibition of the thrombus formation process exerted by LMWH depends on the initial stimulus of blood coagulation (i.e., intrinsic or TF pathway activation) and on the concentration of TF, assay conditions that are closer to the physiology, such as a minimal TF-triggered whole blood ROTEM, have been proved to better reveal the anticoagulant effect of LMWH. Furthermore, the introduction of other reagent-supported tests, such as the PiCT-ROTEM test modification, could permit rapid and adequate LMWH dose adjustments and identify optimal cut-off values to initiate hemostatic interventions in critically ill and bleeding patients. 

At present, most studies have been conducted in vitro with the use of citrated blood; they do not necessarily reflect in vivo hemostasis and the real effect of in vivo administered LMWH thromboprophylactic doses. It would be interesting in future studies to use fresh whole blood for monitoring LMWHs in vivo with the start of ROTEM analysis very quickly after sampling. Therefore, further development of specific test reagents and validation studies are necessary to optimize the ROTEM device in detecting LMWH doses.

### 3.3. Fondaparinux

Fondaparinux, an injectable selective FXa inhibitor, is a synthetic pentasaccharide that binds to the activation site of antithrombin, thereby increasing its activity towards FXa inactivation 300-folds [7]. It is used for thromboprophylaxis in medical and orthopedic patients for the treatment of VTE and for non-ST elevation acute coronary syndromes as indicated by an OASIS-5 study [85]. 

The drug has high bioavailability (about 100%) with an elimination half-life of 15 to 17 h. Since it is not metabolized and excreted through the kidney, a fondaparinux dose needs to be adjusted in patients with renal failure [86].

Although, based on the structure of heparin, fondaparinux is different from both UFH and LMWH, it has no effect on PT and it has a weak effect on aPTT whose degree of prolongation depends on reagent sensitivity and plasma drug concentrations [6,87]. Usually, laboratory monitoring is based on a measure of anti-FXa levels evaluated 3 h after injection [6]; however, during fondaparinux reversal, anti-FXa levels do not change regardless of the chosen reversal agent, whether it is prothrombin complex concentrate (PCC) or recombinant activated factor VII (rFVIIa) [88]. The thrombin generation test (TGT) is probably superior to traditional coagulation tests [89], but because of its technical complexity, it has not become widespread in routine use. Therefore, all previous tests are not adequate to monitor the anticoagulant effects of fondaparinux. 

To overcome these obstacles, clinicians hypothesized the use of the VETs for fondaparinux anticoagulant activity monitoring. Unfortunately, only limited data are available in the literature regarding the effects of fondaparinux on ROTEM parameters in human blood [90]. Eller et al. [90], using blood samples from healthy volunteers that were spiked in vitro with therapeutic and supratherapeutic concentrations of fondaparinux, found that supratherapeutic concentrations significantly affected INTEM-CT without any significant modification of clot formation in the EXTEM assay. No influence on therapeutic concentrations was observed. Consistent with these observations was a study performed by Godier et al. [91] in anesthetized rabbits with thrombosis induced by an injury on the carotid artery treated with fondaparinux 3 mg/kg (a selected dose to induce bleeding) or saline solution (control group). The authors found that only INTEM coagulation parameters were modified. CT was 2-fold increased and CFT 10-fold increased as compared with the control group. Fondaparinux also slightly, but not significantly, decreased the amplitude at 15 min (A15), MCF, and α angle; it modified none of the EXTEM or FIBTEM parameters. Moreover, the authors demonstrated that PCC corrected the INTEM parameters; CT and CFT were shortened, whereas A15 and α angle were increased by 50% and 40%, respectively, compared with fondaparinux.

In conclusion, based on small literature data, ROTEM analysis focuses on the presence of supratherapeutic concentrations of fondaparinux, but not on therapeutic or prophylactic doses. Moreover, the VET could be useful to monitor its reversal. Further human studies are needed to confirm these limited results.

### 3.4. Direct Thrombin Inhibitors (Argatroban and Bivalirudin)

Argatroban is a parenteral direct thrombin inhibitor approved for alternative anticoagulation of patients with proven or suspected heparin-induced thrombocytopenia (HIT) [92,93]. It reaches its steady-state plasma levels 1–3 h after the start of intravenous administration [94]; it is primarily metabolized in the liver, and its half-life is 45 min [95]. 

aPTT is recommended for routine use, but very low doses often result in prolonged aPTT, and the correlation between aPTT values and plasma argatroban concentration is poor [96,97,98,99,100]. In the high-dose range, ecarin clotting time (ECT) has been proposed [101]; however, this test is only available in specialized hospitals.

The ability of ROTEM to monitor argatroban has been reported with conflicting results in relation to argatroban concentrations [102,103]. Engstrom et al. [102] found a significant and strong correlation between argatroban concentrations and INTEM-CT in ex vivo spikes of blood of healthy volunteers (r = 0.98, *p* < 0.0001). They found a strong correlation between aPTT value and INTEM-CT (r = 0.97, *p* < 0001), especially in the clinically relevant therapeutic range up to 100 sec of aPTT prolongation. Furthermore, INTEM-CT seemed not to be sensitive enough to detect argatroban at low doses. On the other hand, in an in vitro study, Schaden et al. [103] demonstrated a strong correlation between argatroban concentration and ecarin-modified thromboelastometry (ECATEM)- CT (*p* < 0.01, r = 0.94) and a moderate correlation with CFT (r = 0.76, *p* < 0.01) and α (r = −0.74, *p* < 0.01), respectively. In this experiment, ECATEM-CT was significantly correlated to argatroban concentration, also at low levels.

Other studies found an impact of argatroban on the ROTEM parameters. Eller et al. [90] reported a significant prolongation not only in INTEM-CT (*p* < 0.05) but also in EXTEM-CT (*p* < 0.05) after argatroban administration in different concentrations (from 0.75 to 2.25 µg/mL). 

It remains unclear whether lower argatroban concentrations might be detected with INTEM-CT or EXTEM-CT. Beiderlinden et al. [104] confirmed a significant correlation between INTEM-CT and EXTEM-CT with low argatroban plasma concentration. However, the strength of these correlations was only moderate.

Bivalirudin, another parenteral direct thrombin inhibitor, differs from argatroban with respect to the route of elimination. It is predominantly excreted by a nonorgan mechanism (proteolysis), with only a minor (20%) component of renal clearance. In addition, the half-life of bivalirudin is shorter than that of argatroban (25 min) [105]. 

This anticoagulant is approved for use for acute coronary events by the US Food and Drug Administration [105]; it is also employed off-label in the setting of extracorporeal support. ACT and aPTT are used for coagulation monitoring during infusion; they rise in a linear fashion with increasing dose [106]. 

In the literature, a recent study [107] compared ROTEM analysis with plasma-based coagulation assayed to monitor bivalirudin anticoagulation in the setting of pediatric extracorporeal support. The authors found a moderate correlation between INTEM-CT and aPTT results (r = 0.54, *p* < 0.0001) and a strong correlation between aPTT Hepzyme and HEPTEM-CT (r = 0.68, *p* < 0.0001). Therefore, ROTEM analysis appeared to be a valid alternative to laboratory assays.

In conclusion, from the limited literature data, it emerges that a ROTEM device should be considered as an option for monitoring the anticoagulant effects of argatroban, even if laboratory test (i.e., ECT) remains the gold standard.

Further prospective studies are needed to identify the full utility of ROTEM as a single device for argatroban and bivalirudin monitoring in cardiological and extracorporeal settings.

## 4. Oral Anticoagulation: Antivitamin K Antagonists (VKAs) and Direct Oral Anticoagulants (DOACs)

Oral anticoagulants, such as VKAs and DOACs, are widely used in the prevention of stroke in patients with NVAF [108,109] or mechanical heart valves, as well as in the treatment of VTE [110,111]. Recently, DOACs have been introduced in thromboprophylaxis for patients undergoing orthopedic surgery [112]. 

VKAs have been the most widespread anticoagulant in the last 60 years [110], and the introduction of DOACs in the past decade has added complexity to the management of periprocedural anticoagulation in urgent clinical scenarios. 

### 4.1. Antivitamin K Antagonists (VKAs)

Warfarin is the most prescribed VKA [113]. It is an anticoagulant that decreases the activity of the vitamin K-dependent procoagulant factors II, VII, IX, X and of the anticoagulant proteins C, S, and Z [113]. It has a half-life of 36 to 42 h [114]. Warfarin is employed more frequently than acenocoumarol because of its longer half-life, which theoretically provides more stable anticoagulation, avoiding factor VII fluctuations that potentially occur during acenocoumarol treatment (half-life of 10 h) [115].

The anticoagulant effect of warfarin is commonly monitored with PT, expressed as the PT-INR (international normalized ratio) to standardize laboratory results. Two types of PT tests are currently in use: Quick-type PT, which is sensitive to factors VII, V, X, and prothrombin (factor II) and fibrinogen deficiencies, and Owren PT, which is only sensitive to vitamin K-dependent factors VII, X, and prothrombin. Owren PT is mainly used in Nordic countries, Benelux, and Japan, whereas Quick PT is extensively used in the rest of the world [116]. The international normalized ratio (INR) based on either of the two PT variants leads to similar results [117].

Due to a low therapeutic index, the risk of bleeding rises with increasing intensity of anticoagulation, as evaluated by INR [118].

#### 4.1.1. ROTEM Monitoring in VKA Therapy

In the literature, several studies have considered the use of ROTEM in the management of VKA anticoagulated patients [119,120,121,122,123,124,125,126,127] (Appendix A). Schmidt et al. [119] demonstrated in warfarin-treated patients that EXTEM-CT has high sensitivity and specificity to detect elevated Owren PT-INR (>1.2). They found EXTEM-CT to be superior to INTEM-CT. At a cut-off of 75 s for EXTEM-CT, sensitivity to detect Owren PT-INR > 1.2 was 0.89, and specificity was 1.0. Moreover, the same authors in another study [120] observed an excellent correlation between a prolonged lagtime of thrombin generation (r = 0.87), just as INR, and EXTEM-CT prolongation in 84 warfarin-treated patients with primary or recurrent VTE. Similar results were found by Nilsson et al. [121], who demonstrated a correlation between EXTEM-CT and FIBTEM-CT with PT-INR Owren (R 0.66, *p* < 0.001). In contrast, Gudmundsdottir et al. [122] showed only a moderate correlation between Owen’s PT-INR and ROTEM-CT (r = 0.27, NS). Probably, evaluating the influence of vitamin K-dependent coagulation factors (i.e., factors II, VII, and X) using ROTEM, the poor correlation could reflect a discrepancy between the selective reduction in factor VII or IX and II or X, as has been previously suggested based on thrombin generation measurements [123]. Moreover, the authors used a highly diluted thromboplastin in the ROTEM assay as opposed to the undiluted thromboplastin in the PT, which could explain the low grade of correlation.

#### 4.1.2. ROTEM Monitoring in VKA Reversal

Some studies have evaluated ROTEM-guided PCC reversal of warfarin [122,123,124]. Rumph et al. [124] in an in vitro study showed that hemostatic components differently affected the onset (CT) and clot growth (α angle and MCF) depending on their respective procoagulant activity and fibrinogen content. In the model of enzymatic deficiency in warfarin-treated plasma, cryoprecipitate improved all three parameters (CT, angle, and MCF) more than fresh frozen plasma, whereas the effects of PCC and fibrinogen concentrate were more specific to CT or α angle/MCF, respectively. In the same line, Spiezia et al. [125] evaluated the in vitro efficacy of three-factor prothrombin complex concentrate (3F-PCC) and four-factor prothrombin complex concentrate (4F-PCC) in reversing warfarin treatment at different degrees of anticoagulation based on EXTEM data and thrombin generation. The authors observed that all PCCs considered in the study normalized EXTEM-CT at a dosage of 0.5 IU/mL, independently of the degree of anticoagulation. No significant differences were found in EXTEM-CT values when comparing 3F-PCC with 4F-PCC. In the same way, Bonderski et al. [126], in a case report, highlighted the role of ROTEM to guide anticoagulation VKA reversal in a patient with a ventricular assist device at a high risk of thrombotic events; this device facilitated the use of an attenuated dosing strategy of 4F-PCC.

Only one study [127] investigated the correlation between INR values and ROTEM parameters in patients treated with acenocoumarol; moreover, it evaluated the efficacy of ROTEM in identifying the safety coagulation threshold for performing invasive procedures in patients undergoing heart valve replacement on acenocoumarol anticoagulation. Blasi et al. [127] found that an EXTEM-CT ≥ 84 s can predict an INR > 1.5 in 92.9% of the patients, whereas a lower EXTEM-CT can predict an INR value < 1.5 in 100% of the cases.

In conclusion, warfarin affects the initial clotting variables of extrinsically activated ROTEM. Other large clinical trials are needed to have sufficient evidence to replace laboratory parameters (i.e., PT and INR) as a first-choice test for warfarin anticoagulation in urgent clinical situations. However, a ROTEM device can be a useful guide to determine the appropriateness of VKA reversal in patients at risk of thrombotic complications.

### 4.2. Direct Oral Anticoagulants (DOACs)

DOACs are categorized into two main classes: oral direct thrombin inhibitor (i.e., dabigatran) and oral direct FXa inhibitors (i.e., apixaban, rivaroxaban, edoxaban, and betrixaban). 

These agents present several advantages compared with VKAs: a more rapid onset and offset, with a peak action 1 to 3 h after intake and half-lives of 10 to 14 h [128], and few drug and food interactions. Moreover, DOACs’ plasma concentration has been reported not to be dependent on body mass index (BMI) [129]. Unlike VKAs, DOACs are given at fixed doses; they do not require routine coagulation monitoring [130] since their effect quickly ends after interruption and, in either case, can be predicted based on a few easily calculable variables (mainly the time from the last dose take, the type of molecule, age, and the glomerular filtrate) [131]. However, detection of DOACs is urgently needed when drug history is not available (e.g., in emergency situations such as severe or aortic dissections), and monitoring of DOAC effects is recommended under specific conditions (e.g., before thrombolytic therapy, emergency situations while receiving a DOAC, after DOAC reversal).

Standard coagulation test results (i.e., aPTT and PT) are of limited value due to high variability, depending on the reagents employed [132,133]. More sensitive assays for monitoring dabigatran activity include thrombin time (TT), diluted thrombin time (dTT), and ecarin clotting time (ECT), while chromogenic anti-FXa assays are more accurate for the monitoring of oral direct FXa inhibitors [134]. 

Recent guidelines on the management of major bleeding in trauma [8] have suggested a potential utility of the measurement of plasma levels of oral direct anti-FXa agents, such as apixaban, rivaroxaban, or edoxaban, in trauma patients treated or suspected of being treated with one of these agents. However, these nonconventional tests may not be suitable for urgent monitoring due to high costs needed for laboratory calibration and long turnaround times [135]. 

In the last years, liquid chromatography–mass spectrometry (LC–MS) assay has been considered the goal standard for the quantification of the levels of DOACs, especially to detect very low levels [136,137]. However, even this analysis is expensive and requires specialized laboratories, which might discourage its use in emergency situations.

#### 4.2.1. ROTEM Monitoring in DOACs Therapy

Several studies pointed out that thromboelastometric tests were applicable to qualitatively monitor the anticoagulant effect of DOACs [138,139,140,141,142,143,144,145,146,147,148,149,150,151,152,153,154,155,156,157,158,159,160,161,162,163,164] (Appendix A) and the reversibility of their activity [165,166,167,168,169,170,171,172,173,174,175,176,177].

Regarding the detection of dabigatran anticoagulant activity with a ROTEM device, some studies evaluated ROTEM reagents in patients treated with dabigatran compared with healthy blood donors (controls) [138,139,140] or analyzed the correlations between ROTEM results and dabigatran plasma concentration [142,143,144,145,146,147,148]. 

Tsantes et al. [138], using the NATEM assay, found a significant prolongation of CT, CFT, and α angle values in patients with NVAF on 110 mg dabigatran twice daily, as compared with healthy subjects (*p* < 0.001, *p* = 0.006, *p* = 0.016, respectively). Similar results were found by Vedovati et al. [139] using the ECATEM assay; in 10 patients treated with dabigatran, the mean CT, the mean MCF, and the mean CT/CT+ catcher at the peak and trough were significantly higher compared with controls (*p* < 0.001 and *p* < 0.001 for CT, respectively; *p* = 0.045 and *p* = 0.048 for MCF, respectively; and *p* < 0.001 and *p* < 0.001 for CT/CT+ catcher, respectively). ROC curve analysis showed a good accuracy for the CT and CT/CT+ catcher in measuring dabigatran anticoagulant activity. In the same line, Korpallova et al. [140] reported a significant difference in CT for INTEM, EXTEM, and FIBTEM for both trough and peak samples in 11 atrial fibrillation (AF) patients treated with dabigatran compared with controls; furthermore, a noticeable difference in MCF FIBTEM was found. However, when comparing trough (121.7 ± 13.5 ng/mL) and peak samples (181.9 ± 23.6 ng/mL), there was a trend towards prolonged peak CT for all three reagents (INTEM, EXTEM, FIBTEM), but the differences did not reach statistical significance. The authors suggested that these results can be explained either by a low simple size or by the fact that a greater increase in dabigatran levels was needed for a significant prolongation of a CT parameter. 

In the literature, a drug concentration ≥ 30 ng/mL has been proposed as a target for antidote administration against DOACs in serious bleedings [141]. Henskens et al. [142], in 75 samples of patients with NVAF, found that EXTEM-CT was sufficiently sensitive (91%) to detect clinically relevant dabigatran activity (≥30 ng/mL), but not CT-INTEM (52%). Moreover, EXTEM-CT showed a good sensitivity for dabigatran with a moderate linearity (R^2^ = 0.5881). Consistent with these findings, Taune et al. [143] showed, in patients with AF, a strong and linear correlation between ROTEM-CT and dabigatran concentration when activated with the reagents EXTEM (r = 0.92, *p* < 0.01) and FIBTEM (r = 0.93, *p* < 0.01), while with INTEM and low TF, the correlation was weaker (r = 0.72 and r = 0.36, *p* < 0.01, respectively). On the other hand, Eller et al. [90], in an in vitro study, found a significant INTEM-CT prolongation associated with EXTEM-CT after dabigatran incubation both at therapeutic (0.1–0.3 µg/mL) and supratherapeutic (20 µg/mL) concentrations. Another small study [144] including 17 AF patients treated with dabigatran confirmed a significant increase in PT, aPTT, and both INTEM-CT and EXTEM-CT 2 h after administration, corresponding to a mean drug level of 128.6 ng/mL. Similar results were found in a recent ex vivo study [145] that showed a significant correlation between dabigatran plasma levels and both EXTEM-CT (r = 0.765, *p* < 0.001) and INTEM-CT (r = 0.702, *p* < 0.05). The authors concluded that EXTEM-CT facilitates the qualitative dabigatran monitoring and, unlike rivaroxaban, even at low plasma levels. In the same line, Comuth et al. [146] performed a ROTEM analysis in samples spiked with dabigatran from healthy donors and in ex vivo samples from patients treated with dabigatran. They demonstrated that EXTEM and FIBTEM-CT correlated with dabigatran plasma concentrations measured with LC–MS/MS without differences between spikes and patient samples. Lastly, in a recent study, Sokol et al. [147] found that INTEM-CT and INTEM-MCF as well as EXTEM-CT and EXTEM-MCF have a strong correlation with the plasma concentration of dabigatran in a real-life population with AF. 

The correlation with dabigatran concentration varied with the trigger used in the ROTEM assay. Using the ECATEM assay, Körber et al. [148], in 10 patients undergoing total knee or hip arthroplasty, showed a correlation coefficient above 0.75 between ECATEM-CT and dabigatran concentration at 2, 6, and 12 h after ingestion on the third postoperative day. The authors concluded that, in a surgical setting, the ECATEM measurement can detect dabigatran in a wide concentration range (0–305 ng/mL). Notably, Schäfer et al. [149] in an in vitro study demonstrated that ECATEM-CT increased early and significantly with dabigatran at all samples; moreover, ECATEM-CT highly correlated with spikes dabigatran concentrations (r = 0.9985; *p* < 0.001). Taune et al. [150] compared modified thrombin-activated ROTEM (thrombin 2 IU/mL) with standard EXTEM in dabigatran-spiked whole blood samples, since dabigatran acts specifically by inhibiting thrombin by binding to its catalytic site. They demonstrated that low dabigatran levels between 20 and 200 ng/mL were detectable with thrombin-activated CT, but not with EXTEM-CT.

Regarding apixaban, in a small cohort of 20 healthy donors, in which whole blood samples were spiked in vitro with the anticoagulant at five different plasma concentrations, EXTEM-CT and INTEM-CT were poorly impacted by apixaban levels, even at high concentrations [151]. Similarly, Mahamad et al. [152] found no significant correlation between EXTEM-CT and apixaban anti-FXa activity (r = 0.56, *p* = 0.72). Moreover, the findings should be interpreted with caution due to the small simple size with low statistical power. These results were not confirmed by Eller et al. [90], who observed a highly significant prolongation of the CT value in INTEM and in EXTEM using apixaban in supratherapeutic concentrations. At therapeutic concentrations of apixaban, CT was increased in EXTEM-CT at 0.1, 0.2, 0.3, and 0.5 µg/mL, whereas a significant increase in INTEM-CT was detected only with 0.5 µg/mL. In the same line, an in vitro study [153] reported a significant EXTEM-CT prolongation after adding apixaban (200 ng/mL) to blood from healthy donors (357.9 ± 52.8 s vs. 165.6 ± 19.7 s in control patients, *p* < 0.01). These results were confirmed by Korpallova et al. [140], who showed significant differences in trough and peak CT for INTEM, EXTEM, and FIBTEM between apixaban-treated patients and controls. Moreover, the ROTEM assay seemed not to be sensitive enough to distinguish the differences between trough and peak apixaban levels. 

From these studies, it appeared clear that values below 200 ng/mL of inhibitors overlap with those observed in the absence of the drug. This lack of sensitivity could originate from the TF content of the EXTEM reagent. Adelmann et al. [154], in an in vitro experiment, showed that decreasing the TF amount up to a final concentration of 0.35 pmol/L (Low-TF-ROTEM) improved the sensitivity to apixaban; the authors found a strong correlation between apixaban plasma concentrations and Low-TF-ROTEM CT (Spearman’s correlation coefficient = 0.81). 

Pailleret et al. [155] demonstrated that a modified ROTEM, triggered with a low TF (5 pmol/L) and a saturating amount of phospholipid vesicles, allowed the detection of as little as 25 ng/mL of apixaban with at least a 1.6-fold increase in CT (*p* < 0.02). CT was the most relevant parameter to detect the FXa inhibitors, whereas MCF was little influenced, and CFT was affected by at least 100 ng/mL of the drug. Moreover, the authors showed that a lengthening of CT value above 197 s could detect samples containing more than 30 ng/mL of apixaban with good sensitivity and specificity. However, modified ROTEM could not exclude a concentration above 30 ng/mL in 10% of cases. These results were not confirmed in a study on ex vivo samples in which Kyriakou et al. [156] demonstrated, in real-life patients with NVAF, that NATEM-CT and CFT were significantly prolonged in patients with apixaban compared with controls. However, there was no correlation between apixaban plasma concentrations and NATEM parameters.

Other studies evaluated ROTEM analysis performed on blood containing rivaroxaban. Schenk et al. [157], in an ex vivo study in which blood was obtained from healthy volunteers and from patients treated with rivaroxaban (15 or 20 mg), found a significant rivaroxaban-dependent prolongation of EXTEM-CT in a dose-dependent manner (100–700 ng/mL, r = 0.76, *p* < 0.05). These results were partly confirmed by Henskens et al. [142], who demonstrated a good sensitivity of EXTEM-CT (96%) and a moderate sensitivity of INTEM-CT (77%) in detecting rivaroxaban activity. In the same line, Casutt et al. [154] reported a slight increase in EXTEM-CT and INTEM-CT 2.5 h after a dose of 10 mg rivaroxaban was administered to 11 healthy volunteers; the authors found normal INTEM-CT values in 7 out of the 11 patients (63.6%) and EXTEM-CT values in 4 out the 11 patients (36.3%). This confirmed that ROTEM tests are not sensitive to a low dose of rivaroxaban. These results were in accordance with that in Seyve et al. [151], who reported a dose-dependent increase in EXTEM-CT and INTEM-CT. However, EXTEM-CT was more sensitive than INTEM-CT in detecting rivaroxaban, but it was unable to systematically detect low concentration (<50 ng/mL) of the anticoagulant. Similarly, Eller et al. [90] reported a prolonged EXTEM-CT only for supratherapeutic but not for therapeutic plasma concentrations of rivaroxaban. Notably, Chojnowski et al. [159], in an in vivo study, found EXTEM-CT to be more sensitive than INTEM-CT after the administration of 20 mg of rivaroxaban. However, EXTEM-CT was not sensitive enough to measure the residual activity of the anticoagulant drug. Accordingly, Oswald et al. [160] showed an increase in EXTEM-CT and INTEM-CT in orthopedic patients treated with 10 mg rivaroxaban compared with those treated with enoxaparin 4 days after major surgery. Moreover, EXTEM-CT was the only variable to increase to a greater degree. The authors concluded that prolonged CT (the only variable to increase) in the EXTEM assay may be useful for detecting treatment with rivaroxaban.

It is known that the peak concentration of rivaroxaban is reached 2–4 h after oral intake [157]. Some studies have evaluated the correlation between plasma rivaroxaban levels and ROTEM parameters [140,144,145,152,162,163]. Hermann et al. [144], in 15 patients receiving 10 mg daily for prophylaxis of DVT after orthopedic surgery, 2 h after intake with a mean of plasma concentration of 128.6 ng/mL, found no modifications of EXTEM-CT (*p* = 0.652) and INTEM-CT (*p* = 0.725). Assuming an influence of rivaroxaban concentration on ROTEM parameters, Perzborn et al. [162], in an in vitro study, demonstrated that a rivaroxaban concentration between 331 and 1807 ng/mL prolonged CT from 1.8-fold to 3.7-fold over the baseline. Similarly, Klages et al. [145], in samples from 34 patients under rivaroxaban therapy before ingestion and 2 to 3 h afterwards, found a significant positive correlation between EXTEM-CT (r = 0.689, *p* < 0.0001) and INTEM-CT (r = 0.595, *p* < 0.001) tests and rivaroxaban plasma levels; moreover, despite a positive correlation with rivaroxaban, the authors concluded that EXTEM-CT was not suitable for qualitative monitoring, as values were distributed over a wide range within the reference area. These results were consistent with a study by Fontana et al. [159], who found, 3 h after administration of 20 mg rivaroxaban in healthy volunteers, significant mean differences of CT but only a moderate correlation of CT with rivaroxaban plasma levels; moreover, they observed a small (+2.2 mm) but significant increase in EXTEM-MCF and no impact on INTEM-MCF and FIBTEM-MCF. Mahamad et al. [152], in a small sample size, demonstrated a correlation with rivaroxaban activity and EXTEM-CT of 0.86, but this was not significant (*p* = 0.2062). Different results were found by Korpallova et al. [140], who reported the availability of CT to react on therapeutic rivaroxaban levels; unlike other DOACs, EXTEM-CT, INTEM-CT, and FIBTEM-CT were significantly more prolonged in rivaroxaban peak samples than in the through samples.

Standard ROTEM assays present a high variability in CT prolongation; CT can remain within the normal range despite high therapeutic plasma concentrations. Some studies evaluated modified ROTEM assays using low TF concentrations (TFTEM) or ecarin (ECATEM). Schäfer et al. [149], in an in vitro study, found that TFTEM-CT correlated with rivaroxaban spike concentrations (*r* = 0.9363, *p* = 0.006), whereas ECATEM-CT did not. Moreover, the TFTEM/ECATEM-CT ratio was above a value of 2 for all dosages of rivaroxaban, even at low plasma concentrations. In the same line, Vedovati et al. [139], using diluted EXTEM assay, with and without the addition of an anti-FXa catcher, demonstrated a good accuracy of CT, CT/CT+ catcher, and CFT/CFT+ catcher in measuring rivaroxaban activity (0.973, 0.987, and 0.860, respectively). In the other line, Tsantes et al. [140], using NATEM in 20 patients with NVAF treated with rivaroxaban, observed a significant increase in CT (*p* < 0.001), CFT (*p* = 0.001), and α angle (*p* = 0.001) compared with the control group. In Adelmann’s study [154], the modified low TF-activated ROTEM, with a decreasing in TF amount up to a final concentration of 0.35 pmol/L, showed a good sensitivity for rivaroxaban detection. Moreover, the authors observed a strong correlation (r = 0.81) between CT and rivaroxaban plasma concentrations. Similar results were observed by Pailleret et al. [155] using modified ROTEM tests that were triggered with the addition of a saturating amount of phospholipid vesicles (10 µmol/L) and a low amount of TF (5 pm/L in platelet-poor plasma or platelet-rich plasma and adjusted to 2.5 pmol/L in whole blood). The authors showed a 1.4-fold increase in CT value (*p* = 0.02) with 25 ng/mL of rivaroxaban; moreover, MCF was little affected, and at least 100 ng/mL of the drug was required to increase CFT significantly. In contrast, in an in vivo study, Kyriakou et al. [156], despite the greater impact of rivaroxaban on CT in comparison with apixaban, demonstrated no correlation between rivaroxaban plasma levels and NATEM parameters.

Data regarding the use of ROTEM for the evaluation of patients anticoagulated with edoxaban are very limited. Two studies considered the impact of whole blood spikes with edoxaban on the ROTEM assay [151,164]. Seyve et al. [151] reported INTEM-CT to be less sensitive than EXTEM-CT; moreover, EXTEM-CT was above the normal range when the plasma concentrations of edoxaban were about 100 ng/mL. Havrdova et al., [164] in 15 healthy volunteers, analyzed the relationship between viscoelastic parameters and edoxaban plasma concentration after 60 mg of edoxaban oral administration. The authors found changes in EXTEM-CT and FIBTEM-CT. Compared with the baseline, EXTEM-CT was significantly prolonged at 2, 4, 6, and 8 h (*p* < 0.001) and at 24 h (*p* < 0.007) after dose. Unlike in Seyve’s study [151], EXTEM-CT and FIBTEM-CT were prolonged also with low plasma concentrations of edoxaban (after 8 h, the mean plasma concentration was 82 ng/mL).

In conclusion, from analysis of the available literature, DOACs affect the ROTEM parameters with the increase in CT in INTEM and EXTEM assays; this reflects the mechanisms of the action of DOACs on the prolongation of PT and aPTT values during the standard coagulation monitoring. However, even though the mean differences of CT are significant, the correlation of CT with DOAC plasma levels appears to be only moderate. ROTEM parameters are nonspecific to DOACs and insensitive to subtherapeutic plasma levels. Modification of ROTEM trigger using TF or ecarin instead of a standard activator improves the sensitivity and specificity of ROTEM testing, but further large studies are required to confirm the limited data and establish cut-off values to detect the specific anticoagulant drug.

#### 4.2.2. ROTEM Monitoring in DOAC Reversal

The clinical utility of ROTEM monitoring after nonspecific (3-PCC, 4-PCC, activated PCC (aPCC)) and specific (idarucizumab or andexanet α for the reversal of FII and FX inhibitors, respectively) DOAC reversal was assessed mainly in in vitro or ex vivo animal studies and in a few clinical trials. 

In the literature, no clinical studies of patients with dabigatran-associated bleeding receiving a target reversal agent under ROTEM monitoring have been conducted. Honickel et al. [165] showed the effectiveness of ROTEM monitoring in the reversal of the anticoagulant effects of dabigatran in a porcine polytrauma model. Significant decreases in INTEM-CT, EXTEM-CT, and EXTEM-CFT were observed after the administration of PCC 50 IU/kg and 100 IU/kg as compared with PCC 25 IU/kg and control animals with reduction in blood loss and better hemodynamic parameters. Similarly, Grottke et al. [166] demonstrated, in the same animal model, that the addition ex vivo to citrated whole blood samples of PCC (30 and 60 IU/kg), aPCC (30 and 60 IU/kg), rFVIIa (90 and 180 µg/kg), and a specific antibody fragment as antidote (aDabi-Fab) (60 and 120 mg/kg) determined a significant reduction in CT and CFT in EXTEM, overall, with the antidotes; on the contrary, rFVIIa had no significant effects on any of the ROTEM parameters. Similarly, the addition of PCC to plasma containing dabigatran resulted in a significant reduction of EXTEM-CT and FIBTEM-CT, but not INTEM-CT [167]. 

For what concerns trauma-induced coagulopathy (TIC) under dabigatran anticoagulation, Honickel et al. [168] performed an experimental study to evaluate the efficacy of 3-PCC and 4-PCC to reverse coagulopathy in a dabigatran anticoagulation/trauma model in pigs using ROTEM monitoring. The authors found that the combined effects of dabigatran anticoagulation and TIC were reduced by both 3-PCC and 4-PCC administration, with a significant decrease in ROTEM-CT and CFT. In the same study, idarucizumab significantly reversed the effects of dabigatran and coagulopathy on ROTEM-CT and CFT. Moreover, treatment with tranexamic acid plus fibrinogen concentrate post-trauma improved the idarucizumab reversal of CFT, MCF, and maximum velocity.

Two other studies showed the utility of CT in ROTEM for the reversal of dabigatran using idarucizumab as antidote. Akman et al. [169], in a porcine polytrauma model with a mean dabigatran concentration above 300 ng/mL, found a positive strong correlation in plasma dabigatran concentration after both intravenous and intraosseous idarucizumab between EXTEM-CT (Pearson’s correlation coefficient, r = 0.81 and r = 0.92, respectively) and INTEM-CT (Pearson’s correlation coefficient, r = 0.93 and r = 0.83, respectively). Moreover, the authors demonstrated that dabigatran reversal with idarucizumab could be evaluated using INTEM-CT and EXTEM-CT when dabigatran concentration levels were below 900 ng/mL. Recently, Takeshita et al. [170], in an in vitro study, confirmed that ROTEM parameters predicted supratherapeutic dabigatran concentration, but the same tests varied in sensitivity to the residual anticoagulant activity after reversal with idarucizumab. The authors showed that, after a fixed dose of idarucizumab, residual dabigatran activity was undetected on INTEM-CT and EXTEM-CT when dabigatran concentration was above 2000 ng/mL. 

Some in vitro studies [162,171,172,173] evaluating the reversal of rivaroxaban anticoagulation with PCC demonstrated the utility of ROTEM monitoring; however, in the literature only two clinical studies in rivaroxaban anticoagulated patients with associated hemorrhage have been reported [174,175]. Bar et al. [174] demonstrated that ROTEM can detect changes in coagulation parameters caused by topical hemostatic agents (i.e., kaolin-based hemostatic agent and chitosan-based agent) in patients on rivaroxaban presenting to the emergency department. Of the samples treated with a kaolin-based hemostatic agent, 87.5% showed reductions in CT, 100% showed reductions in CFT, and 75% showed increases in MCF; moreover, of the samples treated with a chitosan-based agent, 75% showed reductions in CT, 37.5% showed reductions in CFT, and 62.5% showed increases in MCF. Recently, Schenk et al. [175] conducted the first prospective clinical study investigating the effect of PCC in patients with bleeding events during rivaroxaban treatment; they found that EXTEM-CT correlated best with measured rivaroxaban levels but did not improve after 4-PCC administration. 

Similar in vitro studies using ROTEM analysis were presented for apixaban reversal effects. Escolar et al. [153] demonstrated that prolongations in EXTEM-CT due to apixaban were corrected by the different concentrates with variable efficacies (rFVIIa > aPCC > PCC). In another in vitro study, Martin et al. [176] assessed in the whole blood of 16 healthy volunteers spiked with therapeutic or supratherapeutic apixaban concentrations the ability of two different doses of aPCC, PCC, and rFVIIa to reverse the anticoagulant effects. All three types of hemostatic agents significantly reduced EXTEM-CT. Moreover, the lowest rFVIIa dose was more effective than the highest aPCC dose. rFVIIa failed to improve the fibrin clot structure but hastened the initiation of coagulation, which was demonstrated by a reduction of EXTEM and INTEM-CT. However, increasing reversal anticoagulant concentration produced no further benefit. Different results were found by Dinkelaar et al. [167], who conducted an in vitro study in which plasma and whole blood were spiked with apixaban, and PCC was added to these samples with different concentrations (0, 0.25, 0.5, 1, 2, and 4 IU/mL). The authors demonstrated no effects on ROTEM parameters; in particular, the addition of PCC did not result in a significant change in response in any of the tested parameters, except for FIBTEM-CT. Based on an EXTEM-CT value and standard coagulation tests, Schmidt et al. [177] confirmed, in patients under prophylactic therapy with apixaban, that PCC did not completely reverse the effect of apixaban; rFVII was the most effective reversal agent.

In conclusion, a ROTEM device appears to be helpful in the management of DOAC reversal in urgent clinical situations and in the choice of the most effective reversal agent; in particularly, in patients treated with dabigatran, it could be useful to avoid an inappropriate use of the expensive antagonist idarucizumab.

## 5. Conclusions and Perspectives

There is growing interest to utilize VETs to assess the activities of different anticoagulant drugs and their reversal in urgent clinical settings. Indeed, clinicians need to know quickly whether anticoagulant drugs are present in the blood and their impact on patient hemostasis. 

Knowing that “time is precious,” the ROTEM instrument proved to be an excellent guiding tool in optimizing the coagulation of bleeding patients [8]. The current evidence supporting the applicability of this device for the management of anticoagulated patients in emergency situations, particularly when the drug history is not available, is expanding. Figure 2 summarizes the changes of ROTEM parameters in patients taking parenteral and oral anticoagulants.

ROTEM appeared to be useful in detecting UFH with high sensitivity, and the INTEM/HEPTEM CT ratio proved to be a good predictor of heparin rebound after CPB. 

Regarding LMWHs, currently there are small data supporting the use of ROTEM for their monitoring at prophylactic doses. Nonactivated ROTEM (NATEM in combination with NA-HEPTEM), a sensitive test for the evaluation of the endogenous activation of hemostasis, and specific test modifications (i.e., PiCT-ROTEM and Low-TF-ROTEM), could allow for the identification of LMWHs even at low concentrations (>0.1 anti-Xa IU/mL). However, at present, most studies have been conducted in vitro, and further clinical studies are needed to confirm these promising results.

Finally, the detection and differentiation of oral anticoagulants (dabigatran, direct FXa inhibitors, and VKAs) turn out to be very useful. 

Recently, three decision algorithms based on standard thromboelastometric tests (EXTEM, FIBTEM, INTEM, and HEPTEM) and new tests, such as ECATEM and TFTEM, have been proposed [178]. The authors showed that the accuracy of detection and differentiation of oral anticoagulants was 78% using standard ROTEM tests alone, improved to 94% by the additional use of two new modified ROTEM tests (TFTEM and ECATEM), and finally, increased to 98% by a more complex decision tree algorithm. Therefore, a decision-making support based on this algorithm integrated in the ROTEM device could provide important information during emergency treatment in orally anticoagulated patients. 

Clinical development of specific antidotes for DOACs is in progress. Although not specifically designed for monitoring warfarin and DOACs, in in vitro trials, EXTEM emerged to be promising as an assessment tool for acute reversal of DOACs. Certainly, the ROTEM device appeared to be useful to guide the clinician in the management of dabigatran reversal and to avoid an inappropriate use of expensive idarucizumab.

Future prospective multicenter trials to confirm in clinical practice the results achieved in in vitro studies will be needed.

## Figures and Tables

**Figure 1 jcm-11-01407-f001:**
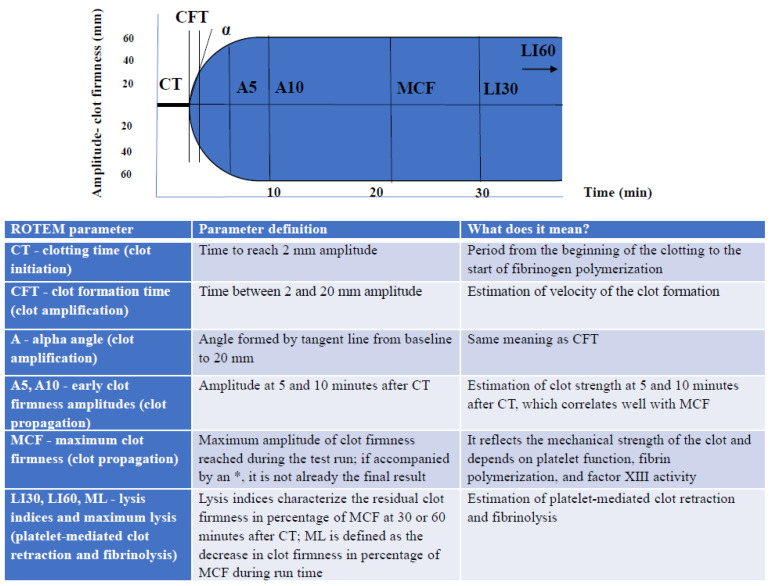
ROTEM parameters and their significance.

**Figure 2 jcm-11-01407-f002:**
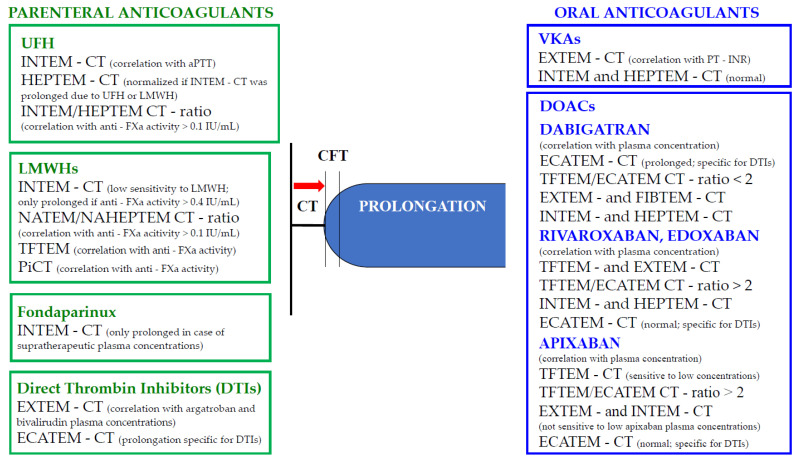
ROTEM parameters in anticoagulated patients with parenteral and oral anticoagulants. Legend: anti-FXa: anti-factor X activated; aPTT: activated partial thromboplastin time; CT: clotting time; CFT: clot formation time; ECATEM: ecarin-activated assay; HFU: unfractioned heparin; INR: international normalized ratio; LMWH: low-molecular-weight heparin; MCF: maximum clot firmness; NATEM: nonactivated thromboelastometry; PiCT: prothrombinase-induced clotting time; PT: prothrombin time; TFTEM: low tissue factor thromboelastometry; UFH: unfractionated heparin; VKAs: antivitamin K antagonists.

**Table 1 jcm-11-01407-t001:** The main characteristics of anticoagulant drugs and their laboratory monitoring.

Type of Drug	Clinical Indications	Route of Administration	Target	Peak Onset	Half-Life Elimination	Standard Coagulation Tests	Specific Coagulation Tests
UFH	VTE (prophylaxis and therapy)	Subcutaneous or intravenous	Factor IIa and Xa	2–4 h	1–2 h	aPTT	Anti-FXa assay
						ACT	
LMWH	VTE (prophylaxis and therapy)	Subcutaneous	Factor Xa (Factor IIa) (molecular weight dependent)	3–5 h	4–5 h	None	Anti-FXa assay
Fondaparinux	VTE (prophylaxis and therapy)	Subcutaneous	Factor Xa	2 h	15–17 h	None	Anti-FXa assay
Argatroban	HIT	Intravenous	Factor IIa	1–3 h	45 min	PT	ECT
						aPTT	
Bivalirudin	PCI	Intravenous	Factor IIa	1–2 h	25 min	PT	ECT
						aPTT	
VKA	NVAF, AHV, VTE	Oral	Factors II, VII, IX, X	36–42 h	5–7 days	PT ↑	None
						INR	
Dabigatran	NVAF, VTE	Oral	Factor IIa	2 h	14–17 h	PT ↑	dTT, ECT
						aPTT ↑↑	
Rivaroxaban	NVAF, VTE	Oral	Factor Xa	2–4 h	7–11 h	PT ↑↑	Anti-FXa assay calibrated
						aPTT ↑	
Apixaban	NVAF, VTE	Oral	Factor Xa	1–4 h	12 h	PT (↑)	Anti-FXa assay calibrated
						aPTT (↑)	
Edoxaban	NVAF, VTE	Oral	Factor Xa	1–2 h	10–14 h	PT (↑)	Anti-FXa assay calibrated
						aPTT ↑	

Legend: ACT: activated clotting time; AHV: artificial heart valves; anti-FXa: anti-factor X activated; aPTT: activated partial thromboplastin time; dTT: diluted thrombin time; ECT: ecarin clotting time; HIT: heparin-induced thrombocytopenia; INR: international normalized ratio; LMWH: low-molecular-weight heparin; NVAF: nonvalvular atrial fibrillation; PCI: percutaneous coronary intervention; PT: prothrombin time; UFH: unfractionated heparin; VKA: vitamin K antagonist; VTE: venous thromboembolism; (↑): normal or slightly increased; ↑: increased; ↑↑: moderately increased.

## Data Availability

No additional data available.

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
