# Peer review of "“In Less than No Time”: Feasibility of Rotational Thromboelastometry to Detect Anticoagulant Drugs Activity and to Guide Reversal Therapy"

_jcm, 2022, doi:10.3390/jcm11051407_

Round 1

Reviewer 1 Report

The paper ““In less than no time”: feasibility of rotational thromboelastometry to detect anticoagulant drugs activity and to guide reversal therapy" by Pavoni et al. is a literary review with the aim of providing most recent data about the role of viscoelastic hemostatic assays such as rotational thromboelastometry (ROTEM) use in the evaluation of anticoagulant levels and the monitoring of reversal therapy.

The article is well written. The study has a good design. The article is logically divided into sections and subsections. The references cited are relevant and adequate, though a bit dated. The work has an average degree of novelty and of good interest to the readers. However, the work is full of data and can be a bit hard to understand, thus the need of one or more figures which can better summarize what reported.

Comments:

  • Line 545: Moreover, DOAC plasma concentrations has been reported not to be dependant to patient BMI (DOI: 1016/j.clinthera.2021.07.003).
  • Line 547: It should be rewritten this paragraph, by underlining the monitoring issue. In fact, monitoring DOACs therapy is not needed since their effect quickly ends after interruption and, in either case, can be predicted based on a few easily calculable variables (mainly the time from the last dose taken, type of molecule, age and the glomerular filtrate). (DOI: 10.3390/medicina55100617).

Author Response

Point 1.

“The article is well written. The study has a good design. The article is logically divided into sections and subsections. The references cited are relevant and adequate, though a bit dated. The work has an average degree of novelty and of good interest to the readers. However, the work is full of data and can be a bit hard to understand, thus the need of one or more figures which can better summarize what reported.” 

Response 1.

We thank the reviewer for the suggestions. We added a figure (Figure 2) which summarized what reported in the text and in the tables.

Point 2. Comments:

Line 545: “Moreover, DOAC plasma concentrations has been reported not to be dependent to patient BMI (DOI: 1016/j.clinthera.2021.07.003)”.

Response 2.

We modified the manuscript adding the suggested reference.

“Moreover, DOACs plasma concentration has been reported not to be dependent to body mass index (BMI) [133].”

Point 3.

Line 547: It should be rewritten this paragraph, by underlining the monitoring issue. In fact, monitoring DOACs therapy is not needed since their effect quickly ends after interruption and, in either case, can be predicted based on a few easily calculable variables (mainly the time from the last dose taken, type of molecule, age and the glomerular filtrate) (DOI: 10.3390/medicina55100617).

Response 3.

We thank the reviewer for this invaluable input. As suggested, the paragraph was rewritten and we underlined the monitoring issue. We added a highly appropriate reference.

“Unlike VKAs, DOACs are given at fixed doses; they do not require routine coagulation monitoring [134], since their effect quickly ends after interruption and, in either case, can be predicted based on a few easily calculable variables (mainly the time from the last dose take, type of molecule, age and the glomerular filtrate) [135]. However, detection of DOACs is urgently needed when drug history is not available (e.g., in emergency situations such as severe trauma or aortic dissections) and monitoring of DOAC effects is recommended under specific conditions (e.g, before thrombolytic therapy, emergency situations while receiving a DOAC, after DOAC reversal.”

Reviewer 2 Report

The authors of the manuscript focused on rotational thromboelastometry a  detect anticoagulant drugs activity.  The authors have come up with an interesting review that is very important for medical research. Rotational thromboelastometry and thromboelastography is a holistic blood coagulation assay.  Rotational thromboelastometry is a viscoelastic hemostatic assay  which has been used in emergencies (trauma and obstetrics), and surgical procedures (cardiac surgery and liver transplants), but experience with this assay in anticoagulant-treated patients is still limited. The manuscript is well structured, but some parts of the manuscript need to be corrected and supplemented in order for this manuscript to be published.

Page 2, lines 59-67: The authors present the clinical practical use of ROTEM in various clinical situations. In this section, the authors should state that it is also used in the diagnosis of bleeding conditions. These data were published in a recent review. This manuscript should be quoted by the authors.: Diagnostics 2021, 11(11), 2140; https://doi.org/10.3390/diagnostics11112140.

Page 7, lines 247-250: The authors state that LMWHs are that they do not require monitoring during the therapy. Yes, this is the correct statement, but there are special clinical preparations for severe coagulopathies, where in perioperative management the patient must be given anticogulation treatment due to the high risk of thrombosis. Perioperative management of hypofibrinogenemic patients is complicated, requiring a specific assessment of the patient's overall hemostasis, taking into account both the bleeding and thrombotic risk. ROTEM can assist in management, this was published in a manuscript that the authors should cite: Thromb Res. 2020 Apr;188:1-4. doi: 10.1016/j.thromres.2020.01.024.

Tables in the text are very clearly written. Authors should consider the better quality of figure 1.

I have to say that with these 178 references. Most of the references are from the last 5 years.

Author Response

The authors would like to thank the Reviewer for his valuable suggestions.

Point 1.

Page 2, lines 59-67: The authors present the clinical practical use of ROTEM in various clinical situations. In this section, the authors should state that it is also used in the diagnosis of bleeding conditions. These data were published in a recent review. This manuscript should be quoted by the authors: Diagnostics 2021, 11(11), 2140; https://doi.org/10.3390/diagnostics11112140.

Response 1.

As suggested, we quoted the suggested review and we added the reference in the section.

“In addition, it has been used in the diagnosis of bleeding conditions due to quantitative and qualitative as well as hereditary and acquired fibrinogen disorders [13]”.

Point 2.

Page 7, lines 247-250: The authors state that LMWHs are that they do not require monitoring during the therapy. Yes, this is the correct statement, but there are special clinical preparations for severe coagulopathies, where in perioperative management the patient must be given anticoagulation treatment due to the high risk of thrombosis. Perioperative management of hypofibrinogenemic patients is complicated, requiring a specific assessment of the patient's overall hemostasis, taking into account both the bleeding and thrombotic risk. ROTEM can assist in management, this was published in a manuscript that the authors should cite: Thromb Res. 2020 Apr;188:1-4. doi: 10.1016/j.thromres.2020.01.024

Response 2.

We thank the reviewer for the feedback and we added the importance of perioperative management of hypofibrinogemic patients with appropriate reference.

“…..or in the perioperative management of hypofibrinogemic patients in which clinicians should take into account both the bleeding and thrombotic risk, monitoring may be required to avoid over or under anticoagulation [57].”

Point 3.

Tables in the text are very clearly written. Authors should consider the better quality of figure 1.

Response 3.

We thank the reviewer for his comments and we have tried to improved the quality of Figure 1. Moreover, we added a figure (Figure 2) to better explain the text and the tables.

Reviewer 3 Report

The review is aimed at analysing the literature on the ABILITY (instead of ‘efficacy’ L.70) of ROTEM, a specific VET with currently two available devices, to detect anticoagulant drugs and to monitor their reversal when invasive procedure and urgent surgery would be more efficiently and safely dealt with thanks to a rapid evaluation of the coagulation system (instead of ‘haemostatic’ L.71).

The bottom line is that ROTEM to some extent has the feasibility of doing so but the clinical usefulness is not really validated – if they are good studies I would have missed, please highlight them in your response, in the abstract and put them in a dedicated section in the text. The case for UFH should be set apart, since INTEM and HEPTEM are long known and widely used tools. The plea for the use of ROTEM during CPB seems very surprising, considering the very use levels of UFH in the blood at that moment – please clarify.

It reads LL.858-9 that “at the present, most of studies have been conducted in vitro and further clinical studies are needed to confirm these promising results.” This should be stated in the abstract.

The authors chose to focus on just one brand of viscoelastometric (and not 'viscoelastic', which relates to the mechanical properties of the forming clot), and this is an important limitation. It should be at least mentioned that there are other brands of VET, with possibilities to address the same issue, but due to the role of the Tem Innovation company in the writing, a closer and comparative analysis is not feasible, which is a pity.

In the title ‘rotational thromboelastometry’ means ROTEM® - the readers should not be confused with a general overview of all available devices from different manufacturers.

Rapid, convenient and reliable detection of anticoagulants in blood is an important issue indeed but the paper 

  • contains parts which are not directly related with the issue
  • does not rely on an explicit methodology regarding the retrieval and analysis of the literature (a systematic review is better than a narrative one)

Main points that are to be carefully considered in a revised version: The abstract and the text should (i) give a hint on why VET could be more sensitive than usual clotting times to DOACs; (ii) address the impact of using whole blood instead of plasma; (iii) be very precise on citrate concentration and other preanalytical aspects, and on accuracy; (iv) highlight the respective properties of the delta and sigma devices regarding the detection of the effect of anticoagulant drugs.

The total required time (“in less than no time”) to get the useful pieces of information from a ROTEM study should be mentioned in the abstract and detailed in the text. Please note that it is possible to speed up the process if the tubes are sent to a lab and that the time advantage is not so conspicuous.

Other important issues that are not addressed: cost and formation of the users; very importantly: influence of haematocrit and platelet count and low fibrinogen (just as it is done for ACT LL.174-5).

The abstract should contain some key points about the very topic.

If the authors are allowed by the editors to be so long, then it is required that what can be found in many text about ROTEM should be much better disclosed, taking advantage of the involvement of the Tem Innovation company in the writing (supposedly substantial – last author’s affiliation).

For instance:

The use of ellagic acid as a contact activator among several possible should be commented upon.

How CT of EXTEM and INTEM compare with PT and aPTT? (would be an indication on the strength of coagulation initiators and influence of blood cells besides plasma)

L.91 What does ‘clot strength’ mean, in the setting of viscoelastic properties?

L.95 on advantages of the new ROTEM version, i.e. sigma, is too vague.

Do clot strands adhere to the pin and cup walls, which are made of what? (L.102)

Why are (L.108) the two non-activated assays no longer available with the sigma version? Why are phospholipids required (L.110) when a clotting test is performed in the presence of platelets?

Regarding what deals with PK-PD of the anticoagulant drugs: very long, with worrisome mistakes. Some examples just below.

Table 1 contains data not really useful for the reading of what deals with the topic, and several errors unfortunately (e.g. (i) LMWH do act thrombin and not only on Xa, and arguably this is their most relevant effect - see Hemker's papers among others; (ii) argatroban is not monitored with PT, and to aPTT is often added a test focused on (meizo)trombin inhibition; (iii) bivalirudin can also be used for HIT). Directly acting drugs are not opposed to heparins and fondaparinux, which act through AT. The tests in the last column on the right are not 'specific', but focused on the target of the anticoagulant - regarding LMWH see above. Regarding parentally administered DTI beside of ECT there are also an ecarin-based chromogenic assays and TT-based assays ('dTT', mentioned only for dabigatran). Finally there are proposals for a 'universal' anti-Xa assay for xabans.

What deals with the PK PD properties of anticoagulant drugs in the text can be shortened. Unfractionated heparin (should be consistently written as such) does not have equivalent activities on Xa and thrombin - purely arbitrary convention (L.153)

Other important points:

LL.848-9 "The current evidence supporting the applicability of this device for the management of anticoagulated patients in emergency situations, particularly when the drug history is not available, is expanding." does not fit with the end of the abstract. 

In the abstract what does 'successfully' add to "the role... has increased"? what is the level of evidence of that? in the conclusion it even reads more emphatically "the ROTEM instrument proved an excellent L.847 guiding tool in optimizing coagulation of bleeding patients." Reformulation required.

 L.860 "...detection and differentiation of oral anticoagulants (dabigatran, direct FXa inhibitors and VKAs) turns out to be very useful.": this is rather optimistic!

Are PiCT-ROTEM and Low-TF-ROTEM formats commercially available? L.142: Please make clear than for PiCT-ROTEM plasma is used, instead of whole blood

From L.178 to L.841: The text is very difficult to read and poorly informative on important points – should thoroughly rearranged, rephrased and shortened. Use the tables by incorporating them into the master file, at least for the most important results. Clearly separate the studies according to their design (in vitro, ex vivo), and focus on studies of good quality, reasonably large cohorts, and with clinical endpoints. Highlight the ROTEM conditions and parameters that seem to be the best ones for each kind of anticoagulant and the modifications that should to be done in the experimental setup to get a better appraisal of anticoagulation and efficacious reversal. In addition an attempt should be made to delineate contradictory results / conclusions and to provide some tentative explanations. The interesting idea to go back to blood not collected into tubes with citrate should be expanded to all anticoagulants, and not just LMWH (L.371), not forgetting to draw the attention of the readers to the caveats associated with such an approach.

The use of the word ‘monitoring’ (e.g. L.563) should be restricted to tests that are proven clinically useful to adjust anticoagulation (UFH and VKA) – otherwise it is infrequent assessment for specific situations, such as urgent invasive procedures.

It would be wise to restrict the review to studies performed with human beings.

More specifically:

L.50: I am not aware of any aPTT reagent with which the CT would be prolonged in the presence of fondaparinux

LL.51-52 the anti-Xa assay is the only one available at the moment, but is far from being a ‘gold standard’. Regarding anti-FXa chromogenic substrate assays L.165 (should be with the plural), the main issue is that they are not equivalent at all.

L.111: what’s the usefulness of mentioning FIBTEM in a text devoted to anticoagulants? is FIBTEM altered by their presence in the blood sample?

L.118: does heparinase have any unwanted effects on the coagulation process beyond heparin degradation? what is its effect on LMWH and danaparoid, if any?

L.119: the ‘cascade’ is not really involved here! The authors should mention meizothrombin

L.122: hirudin is no longer available

L.126: the wording ‘near-physiological conditions’ is debatable; please at least expand

LL.197-199 about detecting heparin with high sensitivity and in excluding residual heparin after CPB: several seemingly contradictions such as ref 37 vs. ref 45. 

L.212: « correctly detected residual heparin in 3 of the only 6 samples showing 212 elevated anti-FXa values after CPB » does not indicate a good sensitivity (50% cf. ref 44 – 45 being the wrong reference I am afraid)

L.263 it reads  "in critically ill patients the presence of antiphospholipid antibodies has been shown to falsely raise anti-FXa levels [64]" I don't see how this is possible in the absence of added phospholipids

L.758: kaolin really? Isn’t it ellagic acid (L.111)?

The rebound phenomenon (UFH) just appears at the end of the paper L.852 without any real explanation in the text (L.230)

Minor

L.90 angle, not area

L.104 and throughout the manuscript: the words ‘thromboelastometric’ and ‘temogram’ are specific to ROTEM and not scientific ones

L.111 ‘extem’ should be in capitals, as throughout the manuscript.

L.119 what is the value of starting the sentence with ‘in particular’?

L.172: gold standard really? what else could it be?

L.184 ‘changeD’?

L.188: found / reported instead of ‘showed’

L.351: what is a ‘heparinase-dependent’ test?

L.364: a test such as ROTEM cannot ‘reveal the antithrombotic effect’ of an anticoagulant

L.377: FXa inhibitor

L.566 what does ‘eventual’ mean here?

L.654: 5pM or 5 pmol/L

References to be carefully checked, for instance

#63 Alhenc-Gelas

#76 ‘Vad’

Author Response

Reviewer 3, comment #1: The review is aimed at analyzing the literature on the ABILITY (instead of ‘efficacy’ L.70) of ROTEM, a specific VET with currently two available devices, to detect anticoagulant drugs and to monitor their reversal when invasive procedure and urgent surgery would be more efficiently and safely dealt with thanks to a rapid evaluation of the coagulation system (instead of ‘haemostatic’ L.71).

Response #1: We changed to wording to: “The purpose of this narrative review is to analyze the literature on the ability of ROTEM to detect different anticoagulant drugs and to monitor their effects and reversal when invasive procedures or urgent surgery would be more efficiently and safely dealt with due to a rapid evaluation of the coagulation system.”

Reviewer 3, comment #2: The bottom line is that ROTEM to some extent has the feasibility of doing so but the clinical usefulness is not really validated – if they are good studies I would have missed, please highlight them in your response, in the abstract and put them in a dedicated section in the text. The case for UFH should be set apart, since INTEM and HEPTEM are long known and widely used tools. The plea for the use of ROTEM during CPB seems very surprising, considering the very use levels of UFH in the blood at that moment – please clarify.

Response #2: Heparin neutralization in ROTEM assays in blood samples with high heparin concentrations as achieved during CPB has extensively been validated in observational (e.g., by Mace et al. 2016 in more than 1,000 patients) and used interventional studies (e.g., by Karkouti et al. 2016 in more than 7,000 patients). Here, EXTEM, FIBTEM and APTEM can neutralize up to 5 IU/mL UFH due to polybrene included in these assays and up to 7 IU/mL in HEPTEM and NAHEPTEM due to heparinase included in these assays. INTEM and NATEM due not include a heparin inhibitor since heparin effects should be detected by these assays.

Gertler R, Wiesner G, Tassani-Prell P, Braun SL, Martin K. Are the point-of-care diagnostics MULTIPLATE and ROTEM valid in the setting of high concentrations of heparin and its reversal with protamine? J Cardiothorac Vasc Anesth. 2011 Dec;25(6):981-6.

Görlinger K, Dirkmann D, Hanke AA, Kamler M, Kottenberg E, Thielmann M, Jakob H, Peters J. First-line therapy with coagulation factor concentrates combined with point-of-care coagulation testing is associated with decreased allogeneic blood transfusion in cardiovascular surgery: a retrospective, single-center cohort study. Anesthesiology. 2011 Dec;115(6):1179-91.

Dirkmann D, Görlinger K, Dusse F, Kottenberg E, Peters J. Early thromboelastometric variables reliably predict maximum clot firmness in patients undergoing cardiac surgery: a step towards earlier decision making. Acta Anaesthesiol Scand. 2013 May;57(5):594-603.

Weber CF, Görlinger K, Meininger D, Herrmann E, Bingold T, Moritz A, Cohn LH, Zacharowski K. Point-of-care testing: a prospective, randomized clinical trial of efficacy in coagulopathic cardiac surgery patients. Anesthesiology. 2012 Sep;117(3):531-47.

Romlin BS, Wåhlander H, Synnergren M, Baghaei F, Jeppsson A. Earlier detection of coagulopathy with thromboelastometry during pediatric cardiac surgery: a prospective observational study. Paediatr Anaesth. 2013 Mar;23(3):222-7.

Gronchi F, Perret A, Ferrari E, Marcucci CM, Flèche J, Crosset M, Schoettker P, Marcucci C. Validation of rotational thromboelastometry during cardiopulmonary bypass: A prospective, observational in-vivo study. Eur J Anaesthesiol. 2014 Feb;31(2):68-75.

Ji SM, Kim SH, Nam JS, Yun HJ, Choi JH, Lee EH, Choi IC. Predictive value of rotational thromboelastometry during cardiopulmonary bypass for thrombocytopenia and hypofibrinogenemia after weaning of cardiopulmonary bypass. Korean J Anesthesiol. 2015 Jun;68(3):241-8.

Karkouti K, McCluskey SA, Callum J, Freedman J, Selby R, Timoumi T, Roy D, Rao V. Evaluation of a novel transfusion algorithm employing point-of-care coagulation assays in cardiac surgery: a retrospective cohort study with interrupted time-series analysis. Anesthesiology. 2015 Mar;122(3):560-70.

Mace H, Lightfoot N, McCluskey S, Selby R, Roy D, Timoumi T, Karkouti K. Validity of Thromboelastometry for Rapid Assessment of Fibrinogen Levels in Heparinized Samples During Cardiac Surgery: A Retrospective, Single-center, Observational Study. J Cardiothorac Vasc Anesth. 2016 Jan;30(1):90-5.

Karkouti K, Callum J, Wijeysundera DN, Rao V, Crowther M, Grocott HP, Pinto R, Scales DC; TACS Investigators. Point-of-Care Hemostatic Testing in Cardiac Surgery: A Stepped-Wedge Clustered Randomized Controlled Trial. Circulation. 2016 Oct 18;134(16):1152-1162.

Scott JP, Niebler RA, Stuth EAE, Newman DK, Tweddell JS, Bercovitz RS, Benson DW, Cole R, Simpson PM, Yan K, Woods RK. Rotational Thromboelastometry Rapidly Predicts Thrombocytopenia and Hypofibrinogenemia During Neonatal Cardiopulmonary Bypass. World J Pediatr Congenit Heart Surg. 2018 Jul;9(4):424-433.

Siemens K, Hunt BJ, Harris J, Nyman AG, Parmar K, Tibby SM. Individualized, Intraoperative Dosing of Fibrinogen Concentrate for the Prevention of Bleeding in Neonatal and Infant Cardiac Surgery Using Cardiopulmonary Bypass (FIBCON): A Phase 1b/2a Randomized Controlled Trial. Circ Cardiovasc Interv. 2020 Dec;13(12):e009465.

Reviewer 3, comment #3: It reads LL.858-9 that “at the present, most of studies have been conducted in vitro and further clinical studies are needed to confirm these promising results.” This should be stated in the abstract.

Response #3: This refers to L871f. Monitoring of LMWH effect by ROTEM has not even be mentioned in the abstract. Therefore, we think that mentioning in the abstract that most ROTEM studies evaluating the effect of LMWH on ROTEM results have been in vitro studies in the abstract is not reasonable due to the limited word count in the abstract.

Reviewer 3, comment #4: The authors chose to focus on just one brand of viscoelastometric (and not 'viscoelastic', which relates to the mechanical properties of the forming clot), and this is an important limitation. It should be at least mentioned that there are other brands of VET, with possibilities to address the same issue, but due to the role of the Tem Innovation company in the writing, a closer and comparative analysis is not feasible, which is a pity.

Comment #4: It is clearly stated in the title that this review paper focus on the feasibility of rotational thromboelastometry to detect anticoagulant drugs and not of viscoelastic devices in general. This would extend the scope of the review, significantly. The difference in detecting anticoagulant drugs is mainly dependent of the reagents used rather than the devices.

Reviewer 3, comment #5: In the title ‘rotational thromboelastometry’ means ROTEM® - the readers should not be confused with a general overview of all available devices from different manufacturers.

Response #5: Rotational thromboelastometry technology is used in the ROTEM delta, ROTEM sigma and ClotPro device. Since ClotPro is using reagents similar to ROTEM delta and sigma, the diagnostic performance of ClotPro may be similar to ROTEM delta and ROTEM sigma but data are still sparse for the ClotPro device.

Furthermore, we did not write “viscoelastic tests” in general, which could lead to confusion in the readers, but rotational (RO) thromboelastometry (TEM) which in the literature identifies only the ROTEM device. While, TEG is an abbreviation of Thrombo (T) elastography (EG). Even for the readers with less experienced in the viscoelastic tests, it is not possible to confuse.

Reviewer 3, comment #6: Rapid, convenient and reliable detection of anticoagulants in blood is an important issue indeed but the paper 

  • contains parts which are not directly related with the issue
  • does not rely on an explicit methodology regarding the retrieval and analysis of the literature (a systematic review is better than a narrative one)

Comment #6: We agree that a systematic review has advantages, but we did not claim to provide a systematic review. In contrast, we clearly state in L71 that this is a narrative review.

Reviewer 3, comment #7: Main points that are to be carefully considered in a revised version: The abstract and the text should (i) give a hint on why VET could be more sensitive than usual clotting times to DOACs; (ii) address the impact of using whole blood instead of plasma; (iii) be very precise on citrate concentration and other preanalytical aspects, and on accuracy; (iv) highlight the respective properties of the delta and sigma devices regarding the detection of the effect of anticoagulant drugs.

Response #7: General advantages and limitations of VET have already been discussed and published elsewhere and would go beyond the scope of this narrative review. We have been requested to revise the manuscript based on the comments provided by reviewer 1 and 2 and submitted the revised version addressing all comments from reviewer 1 and 2 in time before the comments from reviewer 3 have been forwarded to us. Therefore, we don’t think that rewriting the whole manuscript as requested by reviewer 3 is not appropriate and not in agreement with the comments provided by reviewer 1 and 2.

Reviewer 3, comment #8: The total required time (“in less than no time”) to get the useful pieces of information from a ROTEM study should be mentioned in the abstract and detailed in the text. Please note that it is possible to speed up the process if the tubes are sent to a lab and that the time advantage is not so conspicuous.

Response #8: We disagree that sending the tubes to a lab for VET analysis does not speed up the process and making results available earlier for clinicians. The opposite has been reported in clinical studies (e.g., Haas et al. 2012). However, the comparison between performing rotational thromboelastometry in the lab or at the point of care was not the scope of this narrative review.

Haas T, Spielmann N, Mauch J, Speer O, Schmugge M, Weiss M. Reproducibility of thrombelastometry (ROTEM): point-of-care versus hospital laboratory performance. Scand J Clin Lab Invest. 2012 Jul;72(4):313-7.

Reviewer 3, comment #9: Other important issues that are not addressed: cost and formation of the users; very importantly: influence of haematocrit and platelet count and low fibrinogen (just as it is done for ACT LL.174-5).

Response #9: Again, this is not an economic analysis and hematocrit, platelet count and fibrinogen plasma concentration mainly effect clot firmness parameter whereas anticoagulants mainly effect clotting times.

Reviewer 3, comment #10: The abstract should contain some key points about the very topic.

Response #10: As typical for a narrative review, we keep the abstract as short as possible which is I agreement with reviewer 1 and 2. In contrast, reviewer 3 requests a lot of additional information in the abstract. We cannot follow both. Therefore, we ask the editor for advice whether we should extend the size of the abstract significantly or not. 

Reviewer 3, comment #11: If the authors are allowed by the editors to be so long, then it is required that what can be found in many text about ROTEM should be much better disclosed, taking advantage of the involvement of the Tem Innovation company in the writing (supposedly substantial – last author’s affiliation).

For instance:

The use of ellagic acid as a contact activator among several possible should be commented upon.

How CT of EXTEM and INTEM compare with PT and aPTT? (would be an indication on the strength of coagulation initiators and influence of blood cells besides plasma)

Response #11: The COI of KG is clearly addressed in the manuscript (L19 and L912f). Furthermore, this is not a method comparison study comparing different activators of the intrinsic pathway in different VET devices (ellagic acid and kaolin) or a study looking at the correlation between ROTEM and standard coagulation test results. This has already been addressed and reported in multiple studies.

Reviewer 3, comment #12: L.91 What does ‘clot strength’ mean, in the setting of viscoelastic properties?

Response #12: “Clot strength” is a term used regularly in the setting of viscoelastic testing (Advanced PubMed Search (thromboelastometry OR thromboelastography OR viscoelastic) AND ("clot strength") = 570 results) and is characterized in ROTEM by the clot firmness parameter A5, A10, A20 and MCF.

Reviewer 3, comment #13: L.95 on advantages of the new ROTEM version, i.e. sigma, is too vague.

Response #13: A method comparison between ROTEM delta and sigma is not the scope of this review paper.

Reviewer 3, comment #14: Do clot strands adhere to the pin and cup walls, which are made of what? (L.102)

Response #14: Clot strands are made of fibrin and platelets. We considered this is as common knowledge. Or do you want to know the material pins and cup are made of?

Reviewer 3, comment #15: Why are (L.108) the two non-activated assays no longer available with the sigma version? Why are phospholipids required (L.110) when a clotting test is performed in the presence of platelets?

Response #15: Only a limited number of assays can be included in a ROTEM sigma cartridge (or any cartridge of other viscoelastic testing devices such as TEG 6s or Quantra). Future cartridges may include other test combinations including non-activated assays. Phospholipids have a significant impact on coagulation times and are included in viscoelastic testing assays to provide better reproducibility of results. However, this is also not the scope of this narrative review paper.

Reviewer 3, comment #16: Regarding what deals with PK-PD of the anticoagulant drugs: very long, with worrisome mistakes. Some examples just below.

Table 1 contains data not really useful for the reading of what deals with the topic, and several errors unfortunately (e.g. (i) LMWH do act thrombin and not only on Xa, and arguably this is their most relevant effect - see Hemker's papers among others; (ii) argatroban is not monitored with PT, and to aPTT is often added a test focused on (meizo)trombin inhibition; (iii) bivalirudin can also be used for HIT). Directly acting drugs are not opposed to heparins and fondaparinux, which act through AT. The tests in the last column on the right are not 'specific', but focused on the target of the anticoagulant - regarding LMWH see above. Regarding parentally administered DTI beside of ECT there are also an ecarin-based chromogenic assays and TT-based assays ('dTT', mentioned only for dabigatran). Finally there are proposals for a 'universal' anti-Xa assay for xabans.

What deals with the PK PD properties of anticoagulant drugs in the text can be shortened. Unfractionated heparin (should be consistently written as such) does not have equivalent activities on Xa and thrombin - purely arbitrary convention (L.153)

Response #16: It is known that LMWHs with longer fragments, such as tinzaparin, inhibits more strongly FIIa, while LMWHs with shorter fragments, such as enoxaparin, exert more specific inhibition of FXa (anti thrombin effect is not the most relevant effect). We have considered the LMWHs with shorter fragments such enoxaparin (used more frequently in the clinical practice). However, to complete the table we added  (Factor IIa) (molecular weight dependent).

Even if aPTT is recommended for the monitoring of argatroban by the manufacturer, PT shows a stronger correlation to argatroban and bivalirudin plasma concentrations compared to aPTT. However, both PT and aPTT only provide a weak correlation to argatroban and bivalirudin plasma concentrations compared to ecarin-based assays.

Seidel H, Kolde HJ. Monitoring of Argatroban and Lepirudin: What is the Input of Laboratory Values in "Real Life"? Clin Appl Thromb Hemost. 2018 Mar;24(2):287-294. 

Ivandic B, Zorn M. Monitoring of the anticoagulants argatroban and lepirudin: a comparison of laboratory methods. Clin Appl Thromb Hemost. 2011 Oct;17(5):549-55.

Gosselin RC, Dager WE, King JH, Janatpour K, Mahackian K, Larkin EC, Owings JT. Effect of direct thrombin inhibitors, bivalirudin, lepirudin, and argatroban, on prothrombin time and INR values. Am J Clin Pathol. 2004 Apr;121(4):593-9.

Bivalirudin provided good results in patients with HIT type 2 in case series and small studies but has not been approved for the use in patients with HIT type 2 by EMA and FDA in general but only in adult patients undergoing percutaneous coronary interventions and the treatment of adult patients with unstable angina/non-ST segment elevation myocardial infarction (UA/NSTEMI) planned for urgent or early intervention.

Reviewer 3, comment #17: LL.848-9 "The current evidence supporting the applicability of this device for the management of anticoagulated patients in emergency situations, particularly when the drug history is not available, is expanding." does not fit with the end of the abstract. 

Response #17: Again, see response #10.

Reviewer 3, comment #18: In the abstract what does 'successfully' add to "the role... has increased"? what is the level of evidence of that? in the conclusion it even reads more emphatically "the ROTEM instrument proved an excellent L.847 guiding tool in optimizing coagulation of bleeding patients." Reformulation required.

Response #18: This sentence is supported by recent international guidelines recommending the use of viscoelastic testing for monitoring of haemostasis (Spahn DR, et al. Crit Care 2019).

“Coagulation monitoring Recommendation 10: We recommend that routine practice include the early and repeated monitoring of haemostasis, using either a combined traditional laboratory determination [prothrombin time (PT), platelet counts and Clauss fibrinogen level] and/or point-of-care (POC) PT/international normalised ratio (INR) and/or a viscoelastic method (VEM). (Grade 1C).

Reviewer 3, comment #19:  L.860 "...detection and differentiation of oral anticoagulants (dabigatran, direct FXa inhibitors and VKAs) turns out to be very useful.": this is rather optimistic!

Response #19:  We are pleased that the reviewer notes our optimism, but we would like to point out that this sentence is based on recent scientific evidence (see reference 178).

Reviewer 3, comment #20:  Are PiCT-ROTEM and Low-TF-ROTEM formats commercially available? L.142: Please make clear than for PiCT-ROTEM plasma is used, instead of whole blood.

Response #20:  No changes have been made because in the text it is explicit that plasma is used.

Reviewer 3, comment #21: From L.178 to L.841: The text is very difficult to read and poorly informative on important points – should thoroughly rearranged, rephrased and shortened. Use the tables by incorporating them into the master file, at least for the most important results. Clearly separate the studies according to their design (in vitroex vivo), and focus on studies of good quality, reasonably large cohorts, and with clinical endpoints. Highlight the ROTEM conditions and parameters that seem to be the best ones for each kind of anticoagulant and the modifications that should to be done in the experimental setup to get a better appraisal of anticoagulation and efficacious reversal. In addition an attempt should be made to delineate contradictory results / conclusions and to provide some tentative explanations. The interesting idea to go back to blood not collected into tubes with citrate should be expanded to all anticoagulants, and not just LMWH (L.371), not forgetting to draw the attention of the readers to the caveats associated with such an approach.

Response #21:  As suggested by the two previous reviewers, we have summarized the studies expressed in tables in a further figure (Figure 2) which clarifies the concepts explained in the text. 

Reviewer 3, comment #22:The use of the word ‘monitoring’ (e.g. L.563) should be restricted to tests that are proven clinically useful to adjust anticoagulation (UFH and VKA) – otherwise it is infrequent assessment for specific situations, such as urgent invasive procedures.

Response #22:  Our review refers to urgent situations and therefore monitoring appears to be an adequate word. The international guidelines used the word “monitoring” when they recommend the use of viscoelastic method in case of bleeding in trauma patients ( Spahn DR, et al. Crit Care 2019)

Reviewer 3, comment #23: It would be wise to restrict the review to studies performed with human beings.

Response #23:   Our review highlights the state of the art of ROTEM device. There is no doubt that this method has widely entered daily clinical practice and more and more clinical studies are demonstrating its effectiveness. Further post-marketing studies are desirable, as we stated in the conclusions. (“Future prospective multicenter trial to confirm in the clinical practice the results achieved in vitro studies, will be needed”).

Reviewer 3, comment #24: L.50: I am not aware of any aPTT reagent with which the CT would be prolonged in the presence of fondaparinux

Response #24:  This concept should be discussed separately, so we decided to remove “fondaparinux” from the text.

Reviewer 3, comment #25: LL.51-52 the anti-Xa assay is the only one available at the moment, but is far from being a ‘gold standard’. Regarding anti-FXa chromogenic substrate assays L.165 (should be with the plural), the main issue is that they are not equivalent at all.

Response #25:  We changed as follows: “the only one available for their monitoring”. We changed with the plural. We added: “ Furthermore, they are not equivalent to all and they result in increased cost due to need for specialized instrumentation for tests and quality of control”.

Reviewer 3, comment #26: L.111: what’s the usefulness of mentioning FIBTEM in a text devoted to anticoagulants? is FIBTEM altered by their presence in the blood sample?

Response #26:  FIBTEM was explained for completeness in the description of the ROTEM device.

Reviewer 3, comment #27: L.118: does heparinase have any unwanted effects on the coagulation process beyond heparin degradation? what is its effect on LMWH and danaparoid, if any?

Response #27:  The purpose of paragraph number 2 (ROTEM device) is to illustrate the device in order to make more understandable to the reader the argument.

Reviewer 3, comment #28: L.119: the ‘cascade’ is not really involved here! The authors should mention meizothrombin.

Response #28:  We confirm that ECATEM is an assay that uses ecarin to initiate the coagulation cascade at the step of thrombin generation.

Reviewer 3, comment #29: L.122: hirudin is no longer available

Response #29:  We deleted “hirudin”.

Reviewer 3, comment #30: L.126: the wording ‘near-physiological conditions’ is debatable; please at least expand

Response #30:  We suppose that the reviewer refers to NATEM assay. The NATEM mode is of clinical significance since no coagulation activators or reagents is added to the blood sample for analysis. Therefore, the results reflect the true coagulation profile of the patient.

Reviewer 3, comment #31: LL.197-199 about detecting heparin with high sensitivity and in excluding residual heparin after CPB: several seemingly contradictions such as ref 37 vs. ref 45. 

Response #31:  We do not understand what the reviewer is referring to. Ref 37 refers to the results of Prakash's study.

Reviewer 3, comment #32: L.212: « correctly detected residual heparin in 3 of the only 6 samples showing 212 elevated anti-FXa values after CPB » does not indicate a good sensitivity (50% cf. ref 44 – 45 being the wrong reference I am afraid)

Response # 32: The authors thank the reviewer for pointing this out. The correct reference is number 44 (by the same author, Mittermayr). We changed the sentence in the text: “On the contrary, other studies showed a good correlation between INTEM/HEPTEM CT-ratio and heparin values >0.1 IU/mL [44-46]. Particularly, Mittermayr et al. demonstrated that the INTEM/HEPTEM CT-ratio correctly identified 56 of the 58 samples as not containing residual heparin and correctly detected residual heparin in 3 of the only 6 samples showing elevated anti-FXa values (>0.1 IU/mL) after CPB”.

Reviewer 3, comment #33: L.263 it reads  "in critically ill patients the presence of antiphospholipid antibodies has been shown to falsely raise anti-FXa levels [64]" I don't see how this is possible in the absence of added phospholipids

Response #33: In the Artim-Esen’ study (Arthritis Rs Ther 2015) you can read: “…..We then measured the specific effect of the anti-FXa reactive IgG on the enzymatic activity of FXa in a chromogenic substrate assay (Figure 3B). The greatest inhibition of FXa activity was observed with APS-IgG inhibition 9.7 ± 0.89% (mean ± SEM), followed by SLE-IgG (inhibition 7.07 ± 1.28%) whereas HC-IgG gave inhibition of only 2.58 ± 0.6%. These differences were statistically significant; P <0.0001 for APS versus HC, P = 0.0002 for SLE versus HC, and P = 0.0008 for APS versus SLE” and “…. we did demonstrate that both APS- an SLE-IgG directly inhibited specific FXa activity in  chromogenic assay, but the effect was greater for APSIgG”.

We changed the sentence as follows: “Furthermore, in patients with antiphospholipid syndrome, the presence of antiphospholipid antibodies has been shown to falsely raise anti-FXa levels, due to a direct inhibition of specific FXa activity in a chromogenic assay”.

Reviewer 3, comment #34: L.758: kaolin really? Isn’t it ellagic acid (L.111)?

Response #34:  We changed as follows: “activator”.

Reviewer 3, comment #35: The rebound phenomenon (UFH) just appears at the end of the paper L.852 without any real explanation in the text (L.230)

Response #35:  It has been added to the text.

Minor

L.90 angle, not area.

Response: We changed as suggested.

L.104 and throughout the manuscript: the words ‘thromboelastometric’ and ‘temogram’ are specific to ROTEM and not scientific ones

Response: We want to point out the reviewer that the topic of the present review is the use of rotational thromboelastometry (ROTEM) or temogram. Therefore, we used scientific and appropriate names (see literature on argument).

L.111 ‘extem’ should be in capitals, as throughout the manuscript:

Response: We changed as suggested (“EXTEM”).

L.119 what is the value of starting the sentence with ‘in particular’

Response: We thank for the grammar suggestion: we deleted “Particularly”.

L.172: gold standard really? what else could it be?

Response: We have nothing to reply.

L.184 ‘changeD’?

Response: We changed as suggested (“change”).

L.188: found / reported instead of ‘showed’

Response: We changed as suggested (“reported”).

L.351: what is a ‘heparinase-dependent’ test?

Response: ROTEM assay (HEPTEM)

L.364: a test such as ROTEM cannot ‘reveal the antithrombotic effect’ of an anticoagulant

Response: We changed as follows: “anticoagulant effect”.

L.377: FXa inhibitor:

Response: We changed as suggested.

L.566 what does ‘eventual’ mean here?

Response: We deleted “eventual”.

L.654: 5pM or 5 pmol/L:

Response: We changed as follows: 5 pmol/L

References to be carefully checked, for instance

#63 Alhenc-Gelas: we modified “Alheno-Gelas”  to “Alhenc-Gelas”

#76 ‘Vad’: we’re sorry, but we did not change (the name of second author is correct: Henrik Vad).

Submission Date

06 January 2022

Date of this review

20 Jan 2022 18:37:58

Round 2

Reviewer 2 Report

The presented manuscript has been corrected in response to the suggestions. The authors have followed the recommendations of the reviewer. After the revision, the provided data and addition of the results became more clear. I would like to thank the authors for resubmitting the manuscript and explaining the obscure points from the previous version.